# Deterministic full-scenario analysis for maximum credible earthquake hazards

Xiang-Chao Wang[1], Jin-Ting Wang [1] ✉ & Chu-Han Zhang[1]

Great earthquakes are one of the major threats to modern society due to their great destructive power and unpredictability. The maximum credible earthquake (MCE) for a specific fault, i.e., the largest magnitude earthquake that may occur there, has numerous potential scenarios with different source processes, making the future seismic hazard highly uncertain. We propose a full-scenario analysis method to evaluate the MCE hazards with deterministic broadband simulations of numerous scenarios. The full-scenario analysis is achieved by considering all uncertainties of potential future earthquakes with sufficient scenarios. Here we show an application of this method in the seismic hazard analysis for the Xiluodu dam in China by simulating 22,000,000 MCE scenarios in 0–10 Hz. The proposed method can provide arbitrary intensity measures, ground-motion time series, and spatial ground-motion fields for all hazard levels, which enables more realistic and accurate MCE hazard evaluations, and thus has great application potential in earthquake engineering.

Earthquakes are one of the major natural hazards that threaten human society due to their severely destructive nature and the occurrence of uncertainty in both time and space[1]. Great earthquakes occurring near urban areas or major projects like nuclear power stations or high dams can lead to catastrophic damage, such as the 1906 $M_w$ 7.9 San Francisco Earthquake[2] and the 2008 $M_w$ 7.9 Wenchuan earthquake[3]. However, the earthquake hazards are not determined by their magnitude alone, and moderate earthquakes may also cause severe disasters under specific circumstances, e.g., the 1994 $M_w$ 6.7 Northridge earthquake[4] and the 2014 $M_w$ 6.2 Ludian earthquake[5]. How to evaluate the seismic hazard for specific sites accurately is still an essential issue for disaster risk reduction[1]. In the past few decades, seismic hazard analysis methods based on the empirical ground-motion prediction equations (GMPEs), e.g., probabilistic seismic hazard analysis (PSHA) or deterministic seismic hazard analysis (DSHA), have been widely used in the design of major projects and the planning of land-use for the aim of seismic risk mitigation[6,7]. However, the empirical GMPEs are usually constrained by the lack of proper ground-motion records, especially the near-fault records from large earthquakes. They cannot accurately account for the complex rupture process and specific site conditions either[8], and their estimates of seismic hazard may have an unexpected upward bias[9]. Additionally, performance-based seismic design and analysis require ground-motion time series[10], but the empirical GMPEs can only give estimates of the ground-motion intensity measures (IMs).

Deterministic broadband ground-motion simulations have been validated by the records of historical earthquakes and the well-recognized empirical ground-motion prediction models[11-14]. Compared with the empirical GMPEs that are widely used in the traditional seismic hazard analysis methods, physics-based ground-motion simulation methods can comprehensively consider the source mechanism, the propagation path, and the effect of local conditions and thus have vast application prospects in the field of earthquake engineering. Recently, several physics-based seismic hazard analysis approaches have been developed using deterministic ground-motion prediction methods and show great potential to accurately evaluate the seismic hazard[9,15-17]. However, those physics-based seismic hazard analysis approaches are usually constrained in the low-frequency range. In fact, high-frequency ground motions are of great importance in the seismic design of engineering structures. Moreover, the complex physical process of earthquakes is not comprehensively taken into account in these methods for two reasons. First, the number of considered earthquake scenarios in these methods is too small compared to the hundreds of thousands of variables in kinematic source models for generating broadband ground motions. Second, the kinematic parameters of the source process, including the fault location,

[1]Department of Hydraulic Engineering, Tsinghua University, 100084 Beijing, China. ✉e-mail: wangjt@tsinghua.edu.cn

the hypocenter, and the distribution of slip, rupture time, rise time, and rake, are not fully taken into account in these methods. Therefore, the obtained seismic hazard results may not adequately reflect the hazard of potential earthquakes that may attack the target site in the future.

In this study, we are committed to developing a full-scenario seismic hazard analysis method for the maximum credible earthquakes (MCEs) based on fully deterministic broadband ground-motion simulations. Herein, the term MCE refers to the earthquake with the largest magnitude that can reasonably be expected to be generated by a specific source on the basis of seismological and geological evidence[18]. Since there may be several significant active faults with different upper magnitude limits in the vicinity, the target site may have multiple MCEs. Due to the complexity and computational cost of implementation, the proposed method is more suitable for evaluating the seismic hazard of major engineering projects subject to extreme earthquakes occurring in the near field of the target site. Given that the seismic design of major engineering projects, such as high dams and nuclear power plants, particularly emphasizes their safety under extreme seismic conditions, the proposed method is currently intended for evaluating the seismic hazard of major engineering projects under MCEs. The MCE hazards, i.e., the hazards for the target site caused by the largest magnitude earthquakes at all considered active faults, are analyzed using comprehensive MCE scenarios at the significant faults near the target site. The newly proposed multidimension source model is used to describe the complex source process of earthquake scenarios. The multidimension source model, composed of multiscale and self-similar subsources, can generate broadband ground motions while considering specific rupture processes and has been validated by the records of historical earthquakes and the empirical GMPEs[19]. Moreover, the variable space dimension of the seismic source is greatly lowered by applying the self-similarity feature of subsource parameters in the multidimension source model[20]. In this way, the full-scenario analysis for the maximum credible earthquakes can be realized by a holistic consideration of the uncertainties in all kinematic parameters of the source process with comprehensive potential earthquake scenarios. Next, broadband ground motions (0–10 Hz) of up to tens of millions of earthquake scenarios are generated using the adjoint simulation method. Finally, the seismic hazard results of the target site can be obtained by statistically analyzing the IMs of all generated ground motions. In this method, as broadband ground motions at the target site are explicitly generated for all earthquake scenarios, the seismic hazards of arbitrary IMs can be obtained, not just spectral accelerations. Therefore, for specific major engineering structures, the proposed method can provide personalized seismic hazard results by using the optimal IMs of them. Moreover, this method can give earthquake scenarios with the specified seismic hazard levels and the corresponding broadband ground motions, which can be directly used in performance-based seismic design and analysis. The spatial ground-motion field around the target site can also be generated using the earthquake scenarios, which is particularly important in the seismic design for large-span structures such as dams and bridges.

## Results

### Model and setting for analyzing MCE hazard

We apply this method to analyze the MCE hazard of the Xiluodu dam in China. The Xiluodu dam is located in southwest China with a height of 285.5 m, and its reservoir capacity can reach 12.67 billion cubic meters. The dam site is at the junction of the active tectonic blocks with a high seismic risk[21]. Therefore, the seismic safety of this dam, especially its seismic performance subject to destructive earthquakes, e.g., MCEs, is a severe concern in the whole cycle of construction and operation.

To evaluate the seismic safety under MCEs for the Xiluodu arch dam, potential faults where MCEs may occur are identified based on the exploration results of fault structures and historical earthquakes around the target site. There are two important fault structures in the near field of the Xiluodu dam site: the Yaziba fault, with a potential magnitude limit of 7.5, and the Leibo fault, with a potential magnitude limit of 7.0 (Fig. 1). It should be noted that the fault structures in the area have not been well explored and their locations in deep underground are subject to large uncertainties. Therefore, in consideration of the uncertainty of fault location, which is of special importance for the seismic hazard of the near-fault sites, it is assumed that MCEs may occur in the seismic zone within 5 km on both sides of the main fault, and the fault ruptures of MCEs are set up every 1 km in the seismic zone (Supplementary Fig. S2). On this basis, the seismic hazards of the Xiluodu dam site under MCEs are studied based on deterministic broadband ground motion simulations.

The topography in the model domain is considered using the digital elevation model data with a resolution of 30 m. Based on the 3D wave velocity structure of East Asia EARA2014[22–24] and the geological exploration information of the Xiluodu site, the 3D numerical model is established to simulate the ground motions. A well-recognized software package based on the spectral element method[25,26], SPECFEM3D, is used to simulate the seismic wave propagation in the model domain. The minimum shear wave velocity at the surface of the model domain is 2000 m/s, and the shear wave velocity gradually increases to 4400 m/s at depth. To reduce the computational amount, the average spacing of the grid points at the model surface is set to 50 m, and the grid size is doubled at the depth to match the high velocity there. Considering that the spectral element method requires five points for each wavelength of seismic waves when simulating the seismic wave field[27], the effective frequency of the simulated ground motions is up to 10 Hz in this study.

Based on the 3D numerical model, the strain Green's tensors from the Xiluodu dam site to all possible source locations in the potential faults are calculated by three independent numerical simulations, and then the strain Green's tensors are post-processed and stored. In the seismic hazard analysis for the Xiluodu dam site, 22,000,000 scenarios of MCE are generated for the Yaziba fault and the Leibo fault, i.e., 1,000,000 earthquake scenarios are generated for each seismic fault in the potential seismic zones. The Green's functions of subfaults for each seismic fault are calculated using the pre-computed strain Green's tensors. For all earthquake scenarios, the ground motions at the Xiluodu dam site are directly synthesized using the Green's functions of subfaults. Ground-motion IMs, e.g., the spectral accelerations, for all earthquake scenarios are calculated to obtain the seismic hazard results of the target site.

Additionally, it is worth emphasizing that our method is capable of incorporating all uncertainties related to potential future earthquakes. In this study, uncertainties of the fault location, hypocenter, and rupture process of earthquakes are considered in the seismic hazard evaluations for the Xiluodu dam site, while the earthquake magnitude is fixed to the maximum magnitude of surrounding faults for the consideration of MCEs, and fault dips are determined based on fault explorations. However, our method is flexible enough to incorporate other variables of earthquakes, such as earthquake magnitude and fault dip, if necessary.

### MCE hazards produced by the model

The exceedance probability corresponding to the IM is regarded as the seismic hazards for the target site, as presented in Fig. 2. The seismic hazards are obtained by summarizing the spectral accelerations at different periods based on the broadband ground motions of all earthquake scenarios. From Fig. 2, the ground-motion intensity of

earthquake scenarios occurring at the same fault shows significant differences at the Xiluodu dam site. For instance, the Peak Ground Acceleration (PGA) at the exceedance probability of 95% is 0.163 g in the river-parallel direction for the Leibo fault. In contrast, the PGA at the exceedance probability of 5% is 0.443 g, which is about 2.7 times the PGA at the exceedance probability of 95%. Considering that earthquake scenarios occurring at a specific fault have the same magnitude, fault geometry, fault-site distance, and propagation path, the significant variation in the intensity of ground motions indicates that the complex rupture process of earthquakes has a great influence on ground motions at the specific site.

Besides, it can be found that the spectral accelerations of ground motions generated by earthquake scenarios of the Leibo fault are similar to that of the Yaziba fault at the same seismic hazard level. For instance, PGA at the exceedance probability of 50% is 0.268 g in the river-parallel direction for the Yaziba fault, while PGA at the same hazard level is 0.254 g in the river-parallel direction for the Leibo fault. It is surprising that although the MCE magnitude of the Yaziba fault ($M_w$7.5) is obviously larger than that of the Leibo fault ($M_w$7.0), the seismic hazard (represented by spectral accelerations) of the Yaziba fault is comparable to that of the Leibo fault for the Xiluodu dam site. Meanwhile, the MCE hazards for the Xiluodu dam site are further analyzed with four representative IMs (Supplementary Figs. S5 and S6), including the root-mean-square of acceleration ($a_{RMS}$), root-mean-square of velocity ($v_{RMS}$), Arias intensity (Ia), and cumulative absolute velocity (CAV). The four IMs at the river-

parallel direction for the Yaziba seismic zone are obviously higher than those for the Leibo seismic zone, indicating that the seismic hazards represented by different IMs may have significant differences for the same site. Therefore, with the optimal IM of specific engineering structures, this study can provide personalized seismic hazards for them, enabling more accurate and reasonable evaluations of the MCE hazards.

## Earthquake scenarios of the specified hazard level

The seismic hazards of the Xiluodu dam site are obtained by the deterministic full-scenario analysis for MCEs. However, the performance-based seismic design and analysis of engineering structures usually require broadband ground-motion time series. For this issue, the seismic hazard analysis method proposed in this study can also give the earthquake scenarios with specified seismic hazard levels and the corresponding broadband ground motions at the target site. Figure 3 shows the rupture process of the earthquake scenarios at both the Yaziba fault and the Leibo fault corresponding to the 50% exceedance probability of PGA in the river-parallel direction. The corresponding acceleration time series at the Xiluodu dam site is also shown in Fig. 3. Ground motions generated by the earthquake scenario at the Yaziba fault have a long duration, and the river-perpendicular component is stronger than the river-parallel component. Meanwhile, ground motions generated by the earthquake scenario at the Leibo fault are stronger in the river-perpendicular direction. The relative distinction of ground-motion intensity in the two horizontal

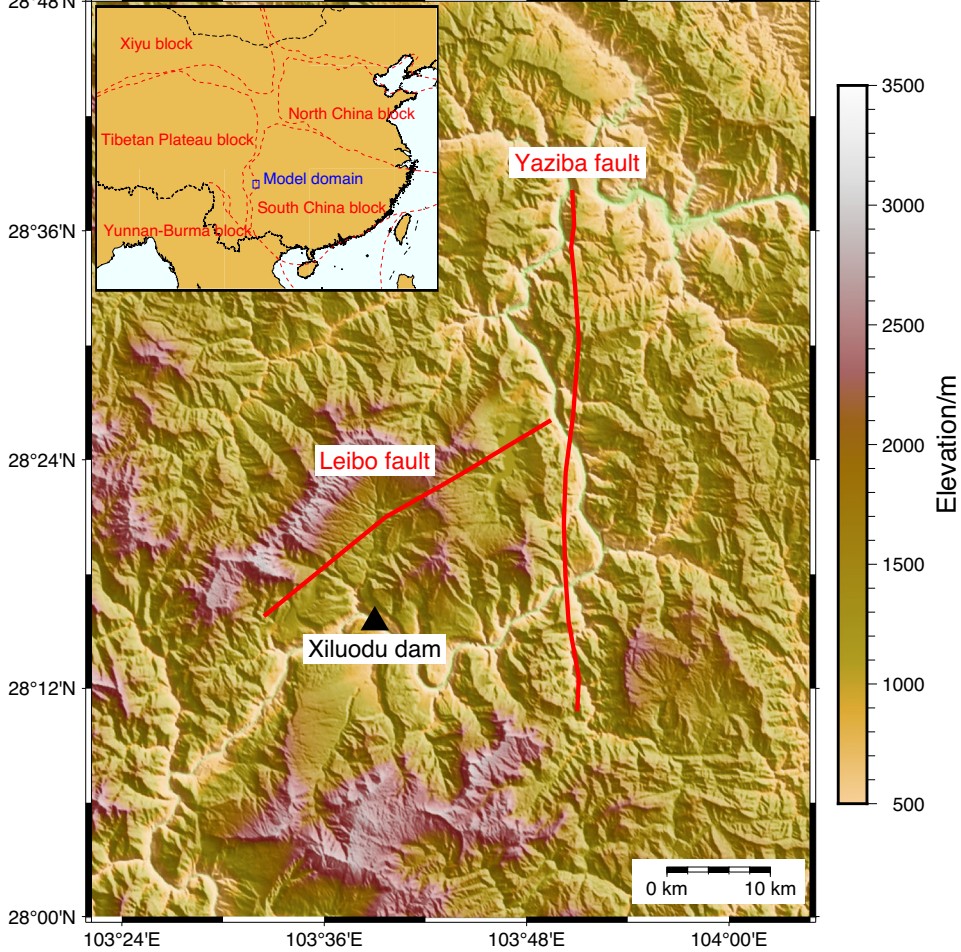

**Fig. 1 | Map of the surrounding region of the Xiluodu dam site.** The black triangle represents the dam site. The red lines indicate the considered faults around the Xiluodu dam site. The Xiluodu dam is located at the junction of the Tibetan Plateau Region and South China Region (Top left inset)[21].

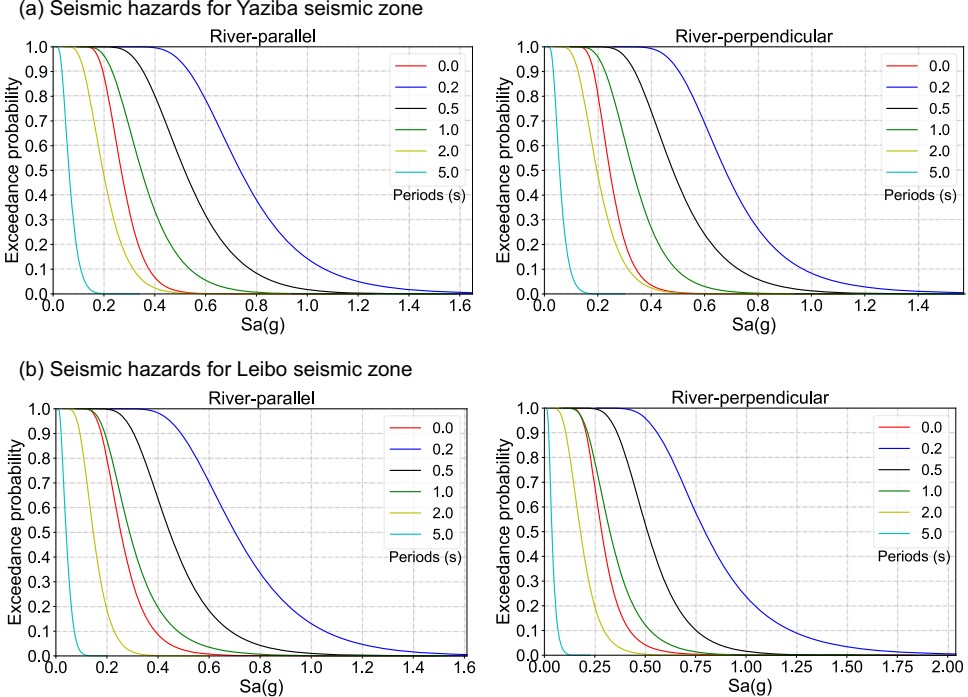

**Fig. 2 | Seismic hazards of the Xiluodu dam site. a** Seismic hazards of the maximum credible earthquake (MCE) at both the river-parallel direction and the river-perpendicular direction for the Yaziba seismic zone. **b** Seismic hazards of MCE at both the river-parallel direction and the river-perpendicular direction for the Leibo seismic zone. Spectral accelerations of all MCE scenarios are summarized to obtain the seismic hazards at six representative periods, i.e., 0.0 s, 0.2 s, 0.5 s, 1.0 s, 2.0 s, and 5.0 s.

directions may be attributed to the differences in source mechanism and fault geometry of earthquake scenarios at the two faults.

Furthermore, by employing the source process of earthquake scenarios corresponding to the specified hazard level, the spatial ground motion field around the target site can be derived by forward simulations. ShakeMaps (PGA of the spatial ground-motion fields) generated by forward simulations for the earthquake scenarios at the Yaziba fault and Leibo fault, as shown in Fig. 3, are presented in Fig. 4. From the ShakeMaps, it is demonstrated that the realistic spatial ground-motion field around the target site can be thoroughly produced by forward simulations using the rupture process of earthquake scenarios corresponding to specific hazard levels. Therefore, for large-span structures (e.g., dams and bridges) that may be significantly affected by the heterogeneous ground-motion field[28], our method can provide more realistic ground-motion input for the seismic design of these structures.

**Comparison with the empirical estimates**

The seismic hazards are compared with four well-recognized empirical ground-motion prediction equations (GMPEs)[29–32]. For comparison purposes, the seismic hazards from our study and the empirical GMPEs are obtained based on the seismic fault located in the center of the seismic zones. As shown in Fig. 5, the seismic hazards in this study are generally in the range of one standard deviation of the four empirical GMPEs, indicating that the proposed method predicates well the spectral accelerations in a wide range of periods. Moreover, the comparison shows that the seismic hazards of this study have much smaller standard deviations of residuals (sigma) than the empirical GMPEs. Hence this study supports the idea that the variability of the seismic hazards at specific sites for certain faults is significantly less than the estimates using the sigma of the empirical GMPEs[9,33].

## Discussion

This study proposes a seismic hazard analysis method based on broadband ground-motion simulations. In this method, the rupture process of earthquakes, the propagation path, and the local site conditions are all determinately considered. Furthermore, based on the self-similar feature of the multidimension source model, MCE scenarios that may attach to the target site are fully analyzed by considering all uncertainties of potential future earthquakes. Nevertheless, the possible nonlinear effects of local sediments during strong earthquakes are not taken into account in this study. Our method may overestimate the MCE hazard to some degree due to ignoring the nonlinear material behaviors[34].

The proposed method is used to evaluate the seismic hazard of the Xiluodu dam in China. We find that the ground-motion intensity of earthquakes occurring at the specific fault is closely related to their physics background. IMs of ground motions generated by earthquake scenarios with the same magnitude and fault location can vary greatly for different rupture processes. Besides, although the MCE magnitude at the Leibo fault ($M_w$7.0) is significantly less than that at the Yaziba fault ($M_w$7.5), earthquake scenarios at the Leibo fault can cause similar seismic risk (represented by the spectral accelerations) to the Xiluodu dam site. It is demonstrated that the seismic hazard of specific sites is closely related to the complex physical background of earthquakes, e.g., the rupture process and propagation path, and cannot be accurately evaluated by simple parameters like magnitude and epicentral distance.

Based on the deterministic broadband ground-motion simulation methods, this study achieves a deterministic full-scenario analysis for potential future earthquakes with a thorough consideration of the physical process of earthquakes. Compared with conventional approaches, this method can much richer results for seismic hazard evaluations, e.g., arbitrary IMs, broadband ground-motion time series, and spatial ground-motion fields for the specified hazard levels,

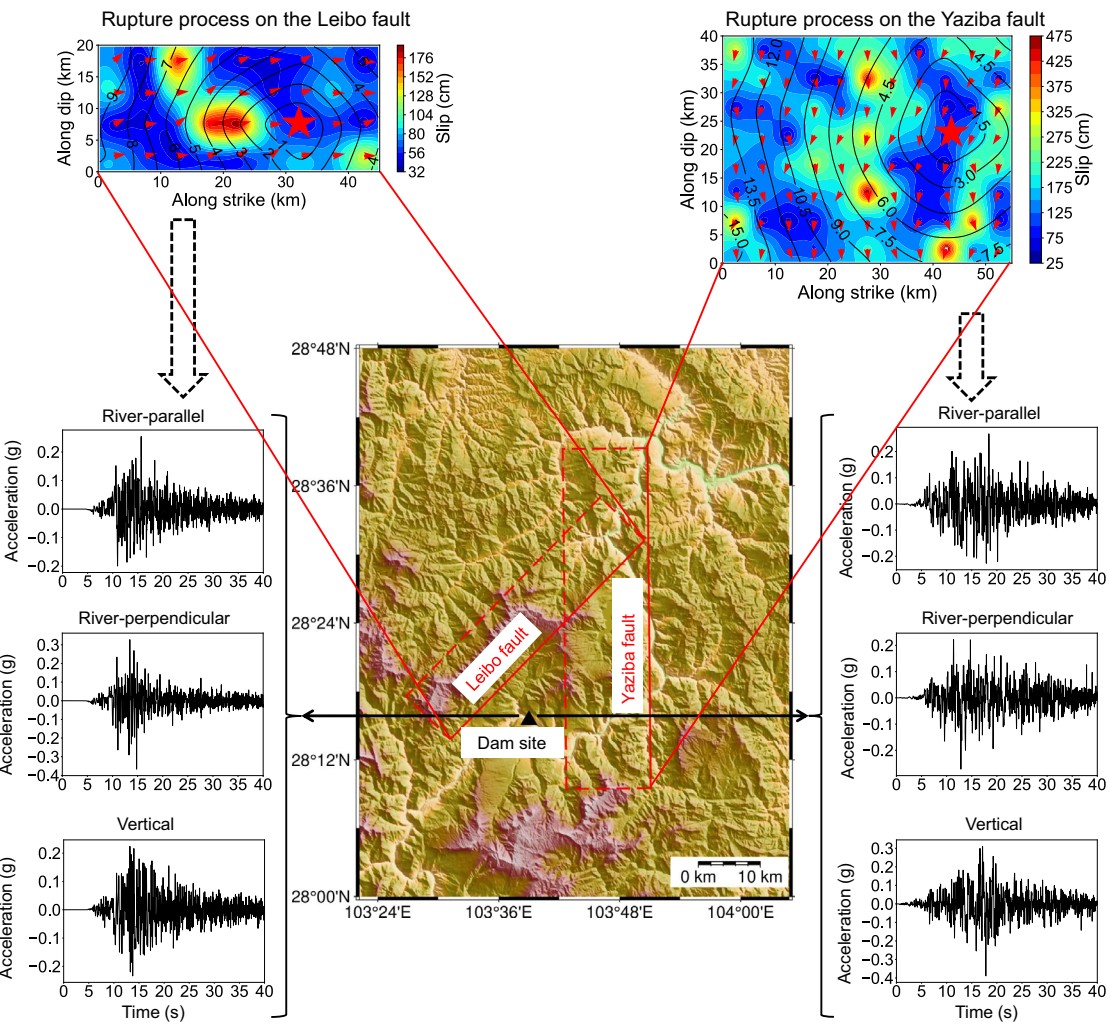

**Fig. 3 | Earthquake scenarios and the corresponding ground motions.** The top edges of the fault planes are indicated by the red solid lines, while the bottom edges are represented by the red dotted lines. Rupture process of the earthquake scenarios corresponding to the 50% exceedance probability of Peak Ground Acceleration (PGA) is presented for both the Yaziba fault and the Leibo fault. The red stars indicate the hypocenter, while the contours, transitioning from red (high values) to blue (low values), are for the slip amount. The black contours represent the rupture time, and the black arrows show the slip direction. Acceleration time series of ground motions at the Xiluodu dam site are presented for the corresponding earthquake scenarios.

enabling more realistic and accurate evaluations of seismic hazards. Furthermore, this study is promising for more ambitious goals to achieve the seismic design fully based on the realistic physical process of earthquakes and holds significant implications in the field of earthquake engineering.

## Methods

### Analysis procedure for MCE hazard

This study proposes a physics-based seismic hazard analysis method toward broadband and full-scenario analysis of the maximum credible earthquakes that may attack the target sites. The proposed method is illustrated in Fig. 6. To begin with, the maximum potential magnitude of future earthquakes that occur in the near field of the target site is determined based on the historical earthquakes and fault structures in the surrounding region. To realize full-scenario analysis of future earthquakes with the maximum potential magnitude, i.e., MCE, the variable space dimension of the rupture process is greatly reduced by applying the self-similarity feature of the multidimension source model. In this way, stable seismic hazards of the target site can be obtained by generating sufficient scenarios of MCE using the Monte Carlo method with a holistic consideration of all kinematic parameters of the rupture process, e.g., the fault

location, the hypocenter, and the distribution of slip, rupture time, rise time, and rake angle.

In the seismic hazards analysis procedure, the target site is considered as a specific location. To deal with the massive computation amount for generating broadband ground motions of all earthquake scenarios, the adjoint simulation method is first used to calculate the strain Green's tensors from the site to all possible source locations. Next, Green's functions of the subfaults of the multidimension source model are obtained by convolving the rupture process on different layers of the subfaults with the corresponding strain Green's tensors. With the pre-obtained Green's functions of subfaults, broadband ground motions of all earthquake scenarios are synthesized directly, and the final seismic hazards of the target site are obtained by statistical analysis of the ground-motion IMs for all earthquake scenarios. Furthermore, after the seismic hazards are obtained, earthquake scenarios corresponding to the specified seismic hazard levels and their broadband ground motions at the target site can also be given and used in the seismic analysis of engineering structures.

### Multidimension source model

To evaluate the seismic hazard of MCE with deterministic broadband ground-motion simulations, the multidimension source model is

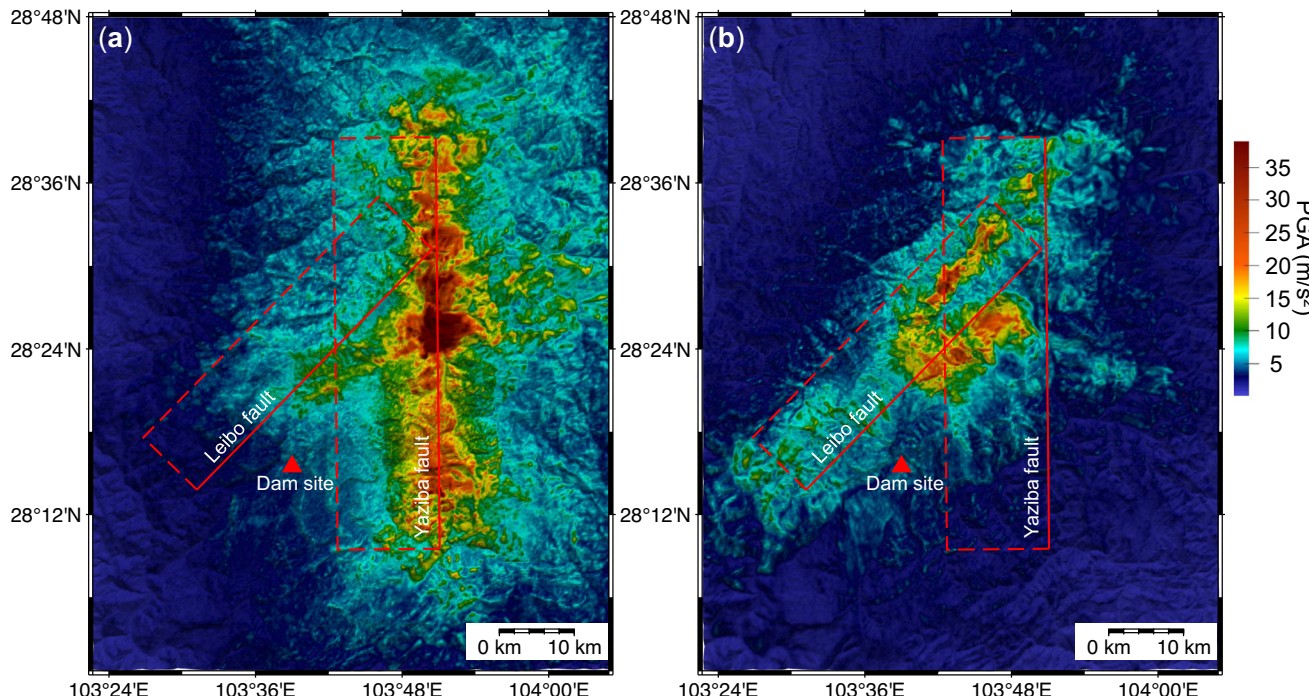

**Fig. 4 | ShakeMaps of the spatial ground-motion fields in the model domain.**
**a** ShakeMap for Peak Ground Acceleration (PGA) of the spatial ground-motion field generated by the earthquake scenario at the Yaziba fault (shown in Fig. 3). **b** ShakeMap for PGA of the spatial ground-motion field generated by the earthquake scenario at the Leibo fault (shown in Fig. 3). The ShakeMaps are generated by forward simulations using the corresponding earthquake scenarios. PGA of the simulated ground motion field in the model domain is presented in different colors. The red triangle represents the Xiluodu dam site. The top edges of the fault planes are indicated by the red solid lines, while their bottom edges are represented by the red dotted lines.

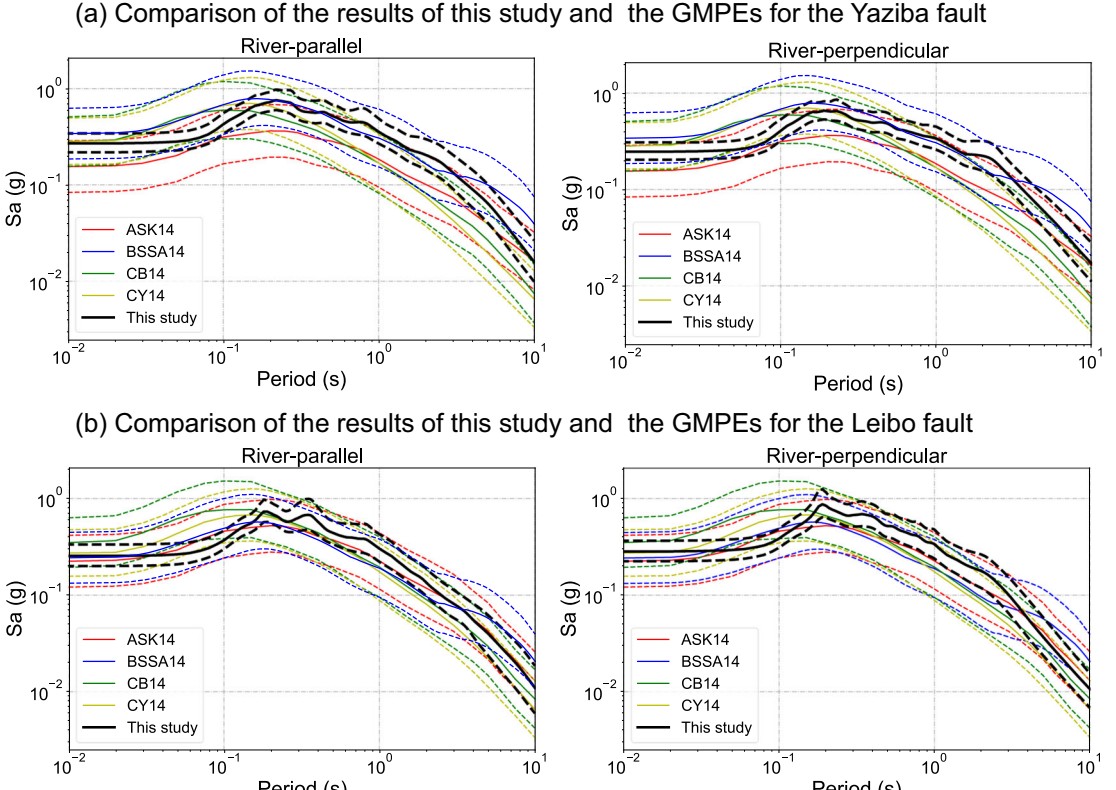

**Fig. 5 | Comparison of the results of this study and four well-recognized empirical GMPEs. a** Comparison of spectral accelerations of this study with the estimates of ASK14[29], BSSA14[30], CB14[31], and CY14[32] for the maximum credible earthquake (MCE) hazards of the Yaziba fault. **b** Comparison of spectral accelerations of this study with the estimates of ASK14[29], BSSA14[30], CB14[31], and CY14[32] for the MCE hazards of the Leibo fault. The solid lines represent the median values, and the dashed lines indicate the range of one standard deviation.

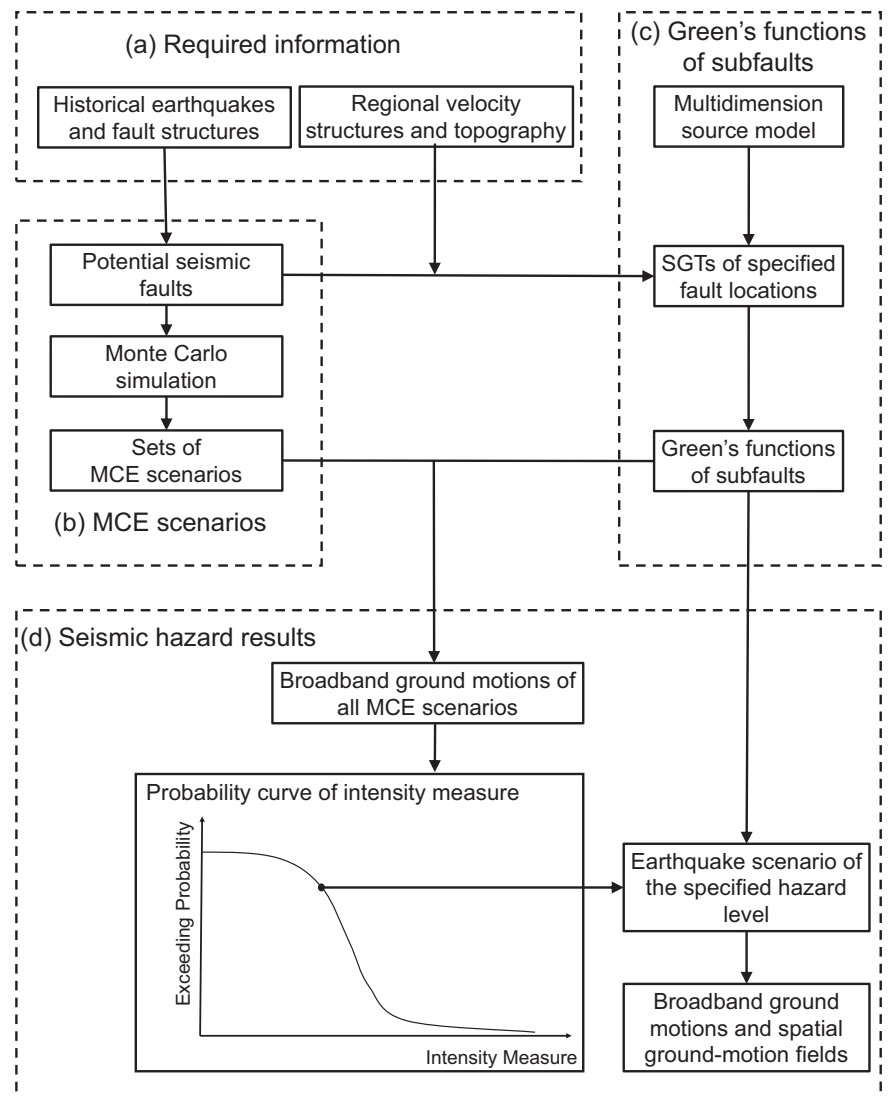

**Fig. 6 | Flow chart of the proposed seismic hazard analysis method. a** Obtaining the required information. **b** Generating the MCE scenarios. **c** Calculating the Green's functions of subfaults. **d** Computing the seismic hazards using broadband ground motions of the MCE scenario sets.

adopted to consider all possible scenarios of MCE. In the multi-dimension source model, the rupture process on the seismic fault is described by several superimposed layers with different subsource scales to generate realistic broadband ground motions, as shown in Fig. 7. For a specified rupture process, the layer number $m$ is determined by the upper limit of the expected frequency contents ($f_{up}$),

$$m = \lceil \log_2(f_{up} T_{min}^{pre}) \rceil \tag{1}$$

where $T_{min}^{pre}$ is the minimum rise time of the specified rupture process. The total seismic moment $M_0$ is assigned to different layers according to

$$M_i = M_0 \frac{R_i}{\sum_{k=1}^{m} R_k} \tag{2}$$

where $R_i$ and $R_k$ is the subsource size for Layer $i$ and Layer $k$, respectively. Besides, the kinematic parameters of subsources, including slip, rupture time, rise time, and rake, on different layers are derived from the specified rupture process. Specifically, the slip, the rupture time, and the rake of subsources on different layers are all consistent with the specified rupture process, while the rise time of subsources on

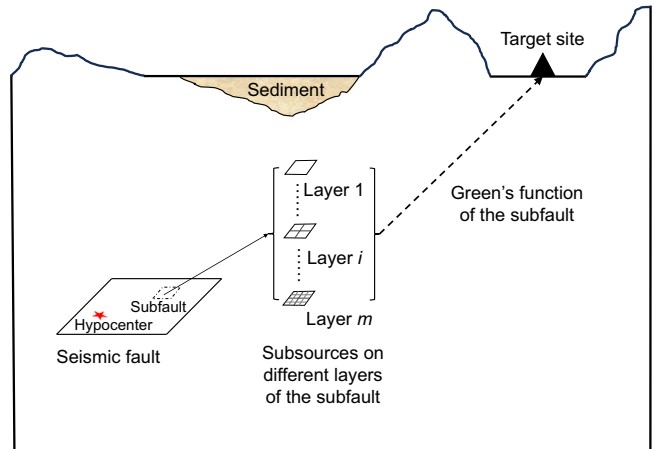

**Fig. 7 | Diagram of the multidimension source model and the Green's function of the subfault.** The source model is divided into several superimposed layers, and each layer is further uniformly divided into subsources with size decreased progressively. The summation of ground motions generated by subsources on different layers of the same local area is regarded as Green's function of the subfault.

different layers is scaled by

$$T_{ij} = \frac{R_i}{R_1} T_j^{pre} \tag{3}$$

where $T_j^{pre}$ is the rise time of the specified rupture process at the same location of the subsource $j$ on Layer $i$.

## Green's function of subfault

To realize full-scenario analysis of MCE, the dimension of variable space of the source process is reduced by applying the self-similar feature of the multidimension source model so that scenarios of MCE can be comprehensively considered with the Monte Carlo method. Specifically, as the kinematic parameters of subsources on different layers of the same area, i.e., the subfault, as shown in Fig. 7, are consistent with the specified rupture process, they can be considered together as a whole in the Monte Carlo method. The summation of subsources on different layers of the same local area, $u_j$, is represented as

$$u_j(t_j, T_j, D_j, \lambda_j) = \sum_{i=1}^{m} \sum_{k=1}^{n_i} G_{jk}^i * S_{jk}^i(t_{jk}^i, T_{jk}^i, D_{jk}^i, \lambda_{jk}^i) \tag{4}$$

where $G_{jk}^i$ is the Green's function between the site and the kth subsource on the jth subfault of the ith layer, $S_{jk}^i$ is the source time function of the kth subsource on the jth subfault of the ith layer, and $t_{jk}^i, T_{jk}^i, D_{jk}^i, \lambda_{jk}^i$ are the rupture time, rise time, slip amount, and rake angle of the kth subsource on the jth subfault of the ith layer, respectively. $u_j$ is used as the Green's function of the jth subfault to generate broadband ground motions of the earthquake scenario. In this way, the number of variables to be considered is significantly reduced, and then the Monte Carlo method is used to generate the scenario sets of MCE by considering all kinematic parameters of the source process.

In this study, to generate the earthquake scenarios with the Monte Carlo method, all kinematic parameters of the source process (including the hypocenter, the distribution of slip, rupture time, rise time, and rake) are randomly generated in certain ranges with necessary constraints. For the source process of earthquake scenarios with, the average slip and rise time are constrained. Specifically, the average slip is determined by the empirical earthquake source-scaling laws[35], and the average rise time is specified according to the empirical relation for the rise time[36]. According to the existing research, the hypocenter of an earthquake may occur at any position of the seismic fault[37,38]. However, due to the very limited number of earthquakes with accurate fault information derived from sufficient studies, as well as the significant randomness of the hypocenter locations, there is no well-recognized conclusion regarding the spatial pattern of epicenters. Therefore, in this study, we simply assume that the hypocenters of earthquake scenarios occur randomly at any position of the seismic fault. Furthermore, it is worth emphasizing that our method is highly adaptable and can be updated to incorporate advancements in the understanding of earthquake nature, such as improved scaling laws for rupture parameters, as well as refined exploration information of faults and regional Earth model. By incorporating these advancements, our method can yield more accurate evaluations of the seismic hazard at the target site. Details of source parameters of the generated earthquake scenarios are provided in the Supplementary Information.

## Ground motions at the target site

After the scenario sets of MCE that may attack the target site are generated, broadband ground motions of these earthquake scenarios at the target site are used to evaluate the seismic hazard there. To solve the expensive computation of simulating many earthquakes, we obtain the ground motions of all earthquake scenarios using the adjoint method by applying the reciprocity of the strain Green's tensors[39–41]. The adjoint methods can generate all needed strain Green's tensors with only three numerical simulations. The Green's function of the subfault $u_j$ is then obtained by convoluting with the pre-obtained strain Green's tensors with the rupture process of all layers of the subfault. Once completed, broadband ground motions at the target site of all considered earthquake scenarios can be easily and quickly synthesized with the Green's functions of subfaults. Broadband ground motions of the MCE scenarios are then statistically analyzed to get the seismic hazards of the target site. By convention, the seismic hazards are presented using the exceedance probability curves of the spectral accelerations. However, as broadband ground motions are explicitly obtained for all considered earthquake scenarios, any IM can be adopted for the seismic hazard analysis in our method.

## Summary of seismic hazard analysis method

Our seismic hazard analysis method starts with some basic seismic information, including the historical earthquakes and the fault structures near the target site, which are used to determine the magnitude and the potential seismic zones of the maximum credible earthquakes. Based on the specified magnitude and the potential seismic zones, the scenario sets of MCE are generated with the Monte Carlo method by considering all kinematic parameters of the rupture process, e.g., the hypocenter, the distribution of slip, rupture time, rise time, and rake on the seismic fault. After the Green's functions of subfaults of the multidimension source model are obtained with the pre-computed strain Green's tensors, broadband ground motions at the target site are directly synthesized for all earthquake scenarios. Finally, the seismic hazards are obtained by statistically analyzing the broadband ground motions of all scenarios of MCE.

## Data availability

The block data used in Fig. 1 is available at https://github.com/gmt-china/china-geospatial-data. The 3D velocity structure of East Asia EARA2014 is available at https://chenseismolab.org/resources/. The digital elevation model data is available at the Geospatial Data Cloud (https://www.gscloud.cn/). Data on the faults and local site information of the Xiluodu dam site is obtained from the non-public seismic safety evaluation report for the project site of Jinsha River Xiluodu Hydropower Station (in Chinese) and is available from the authors upon request.

## Code availability

The code of SPECFEM3D was downloaded from: https://geodynamics.org/cig/software/specfem3d. The empirical GMPEs were calculated using the pygmm software (https://github.com/arkottke). The Generic Mapping Tools for generating some figures are available at https://github.com/GenericMappingTools. Codes for the data processing and analysis are available from the authors upon request.

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

## Acknowledgements

J.T.W. was financially supported by the National Natural Science Foundation of China (No. 52339007), the National Key R&D Program of China (No. 2022YFC3004303), and the State Key Laboratory of Hydroscience and Hydraulic Engineering (No. 2021-KY-04). The authors express their sincerest gratitude for the support.

## Author contributions

X.C.W., J.T.W. and C.H.Z. conceptualized and designed this research. X.C.W. and J.T.W. developed the methodology. X.C.W. performed the analysis and wrote the initial version of the manuscript. All authors contributed to the revision process and provided comments on different versions of the manuscript.

## Competing interests
The authors declare no competing interests.
