## [Peer Review File · Nature Communications]

nature portfolio

Peer Review File

Deterministic full-scenario analysis for maximum credible earthquake hazardsReviewers' comments:

Reviewer #1 (Remarks to the Author):

This manuscript presents results from numerical simulations of ground shaking from two faults in China, randomizing some earthquake source parameters. The study mixes hazard analysis (characterizing earthquake sources and resulting shaking) with adjustments to the method for predicting shaking. The use of earthquake scenarios for hazard analysis, and the use of numerical simulations to predict ground motions, are both areas of research that have been studied for decades. The authors should be clearer about where their intended contribution is, as many of the statements of the paper are largely repeating statements made elsewhere. Without a clear innovation here, the manuscript does not appear suitable for publication. A few minor comments are below as well.

"Maximum credible earthquake" is a discouraged terminology in recent decades. It was previously used to refer to earthquake ground motions before its phasing out, and it is unclear here whether the term is being used to refer to a ground motion or an earthquake magnitude (the full set of earthquake source parameters is a vector and so doesn't have a clear maximum). There is no "maximum" earthquake ground motion either, as the authors acknowledge through their use of millions of stochastic simulations. An alternate terminology would be preferred.

Why generate millions of ground motion simulations, but consider only a single fixed earthquake magnitude, dip, etc. for each fault? Some thought and discussion is needed regarding uncertainties in potential future earthquake scenarios, and how they are reflected in the analysis.

Line 108, 2000 m/s at the surface is very stiff. Is this representing a measured value at the site, or is this high value necessary due to numerical modeling needs? A clarification is needed.

Sincerely,
Jack Baker

Reviewer #2 (Remarks to the Author):

NOTEWORTHY RESULTS

This paper provides a case study of potential ground shaking from earthquakes on two faults close to a major dam in China, demonstrating that the hazard is similar from both faults despite the variations in potential maximum credible earthquakes. In most respects the method here is more noteworthy than the specific results for the case study, using an extremely large number of 'scenarios' to better define uncertainty in ground shaking at a specific site.

SIGNIFICANCE OF THE WORK

The field of earthquake hazard and ground motion simulation is large, and so such a study will no doubt be of interest to a large number of specialists. However, I'm unclear of the wider interest or implications for this work beyond providing slight decreases in output variability from existing methods. Perhaps I missed this in the Introduction section, but it's unclear to me how the suggested approach is driving the science forward beyond incremental gains, and the focus on a very specific case study doesn't help in this regard.

It appears to me that the main significance of the work is the ability to match hazard estimates from existing Ground Motion Prediction Equations albeit with reduced variability/uncertain. While I don't question this is an important step forward, I'm unconvinced this is a substantially significant contribution for a multi-disciplinary audience.

DOES THE WORK SUPPORT THE CONCLUSIONS AND CLAIMS

Partly. The method is able to produce a large number of simulations and produce results that are in keeping with previous findings with reduced variability. However, I'm not sure the approach makes the advances that the authors claim in the Introduction section

ARE THERE ANY FLAWS IN THE DATA ANALYSIS, INTERPRETATION AND CONCLUSIONS?

In the main, this study appears robust, with suitable data analysis and interpretation. Where I am left confused is in the conclusions and how other researchers can and should take this work forward. If this is some major leap forward in the science of earthquake hazard derivation then I must be missing it, and I would recommend the authors make this point more clearly. Perhaps I am under-playing the technical advancements that have gone into achieving these results, but to me this seems like a fairly standard Monte Carlo model using existing assumptions and understanding, rather than generating some new understanding. Even for this particular case study, the results appear to simply confirm the existing knowledge from empirical GMPEs. I would encourage the authors here to make a better 'sales pitch' as to why this approach is so important, and in particular focus on making that clear and obvious to a broader audience – at present I imagine much of the detail is really only relevant for a sub-set of specialists in ground motion modelling.

IS THE METHODOLOGY SOUND? IS THERE ENOUGH DETAIL PROVIDED IN THE METHODS FOR THE WORK TO BE REPRODUCED?

From what I can tell, the method is appropriate and robust, and the paper describes the approach in sufficient detail to allow reproduction. However, the precise details of the approach are slightly beyond my specific expertise and so other referees may have more pertinent comments.

My main concerns here are around the wider applicability and assumptions that go into the model. If the authors envisage this approach being widely adopted, then I would suggest they focus on describing how this method can be applied elsewhere, particularly in regions where there is little information on historical earthquakes and fault structures (i.e. most of the world's fault-lines!). Further, there appears to be no real scrutiny of the relationships linking basic seismic information to the MCE. So, in reality, is the main contribution of this paper the ability to provide a very large number of models using existing understanding and assumptions?

Reviewer #3 (Remarks to the Author):

The paper is interesting and well written and covers an important topic. It deals with the estimation of the MCE at a dam site using physics-based ground motion.

Description

The authors state that the maximum frequency of the simulations is up to 10Hz and that all important variability of the parameters influencing the results is considered.

They selected the faults and assigned the maximum expected magnitude. Running thousands of scenarios by changing source parameters, they computed accelerograms at the edge of a site by extracting relevant intensity measurements.

They evaluated the distribution of the different MIs. They claim that the results express the MCE.

Observations:

The topic is quite interesting as both MCE assessment and the use of physics-based ground motion simulations are active research topics. However, more details about specific input patterns need to be added. Also, authors must state what they mean by MCE. Since it is a seismic input for the structural analysis, it should represent the input that provides the maximum excitation for the structure under analysis. Since the dam is very stiff, it is possible that scenarios with lower magnitudes could excite the dam more than the scenarios considered here. So is it really MCE?

To really find the MCE as an input for the structural analysis, multiple scenarios should be combined, also considering lower magnitudes. A procedure to account for this, also through ground motion simulations, can be found in the reference attached to the end.

The uncertainty in the structural model has not been considered (and this may explain part of the reduced variability with respect to the GMPE) and the distribution of the parameters implemented in the source model is not well described.

In short, the paper presents interesting aspects but lacks a thorough discussion on the possible limitations of the presented results. A minor revision is therefore suggested to clarify the following specific comments.

Specific comments:

- The structural model is not described
- The velocity seems to be too fast to reach the specific conditions of the real site, please look into this
- The maximum frequency is a function of the source, the structural model and the precision of the site specific layers. The high frequencies are strongly influenced by the local soft layers. It seems they are not included in the model
- Line 129: An occurrence model for magnitudes has not been used so it is better not to call the IM distributions "hazard curves".
- Line 155: it is true that accelerograms are available, however since it is not imaginable to use all of them, it is necessary to apply an accurate selection procedure based on the dynamic properties of the structure to be analysed.
- Fig. 4: it is true that the PBGMS could be a tool to obtain more information on seismic hazard and also to limit uncertainties. But in this case the variability appears to be very small. A discussion of this needs

to be added. Uncertainty on soil parameters has also not been included and this would surely increase dispersion (particularly at high frequency)

- Line 210. See previous discussion on “What is MCE?”

- Line 217: At least in the supplementary material it might be helpful to explain the parameter distributions and how that variability was evaluated.

Chieffo, N., Fasan, M., Romanelli, R., Formisano, A., Mochi, G., 2022. “Physics-Based Ground Motion Simulations for the Prediction of the Seismic Vulnerability of Masonry Building Compounds in Mirandola (Italy)”, *Buildings*, 11, 667. <https://doi.org/10.3390/buildings11120667>

Hassan, H.M., Fasan, M., Sayed, M.A., Romanelli, F., ElGabry, M.N, Vaccari, F., Hamed, A., 2020. “Site-specific ground motion modeling for a historical Cairo site as a step towards computation of seismic input at cultural heritage sites”, *Engineering Geology*, Volume 268, 105524, <https://doi.org/10.1016/j.enggeo.2020.105524>

Fasan, M., Hassan, H.M., Magrin, A., Vaccari, F., Romanelli, F., ElGabry, 2023. “Multi-scenario Physics-Based Seismic Hazard Assessment of Cultural Heritage Sites”, in El-Qady, G.M., Margottini, C. (eds) *Sustainable Conservation of UNESCO and Other Heritage Sites Through Proactive Geosciences*. Springer Geology. Springer, Cham. https://doi.org/10.1007/978-3-031-13810-2_26

Response to the comments

The authors would like to thank the three reviewers for their constructive comments, which were of great help in improving the manuscript. The authors have carefully considered the detailed comments, and the point-by-point responses to the reviewers' comments are given below (Reviewers' comments are in *italic fonts*, responses are in normal fonts and changes of the manuscript are in **bold fonts**).

Reviewers' comments:

Reviewer #1 (Remarks to the Author):

Comment 1.1:

This manuscript presents results from numerical simulations of ground shaking from two faults in China, randomizing some earthquake source parameters. The study mixes hazard analysis (characterizing earthquake sources and resulting shaking) with adjustments to the method for predicting shaking. The use of earthquake scenarios for hazard analysis, and the use of numerical simulations to predict ground motions, are both areas of research that have been studied for decades. The authors should be clearer about where their intended contribution is, as many of the statements of the paper are largely repeating statements made elsewhere. Without a clear innovation here, the manuscript does not appear suitable for publication. A few minor comments are below as well.

Response: We appreciate Prof. Backer for carefully reading our manuscript and making the insightful, critical, and constructive feedbacks. We agree that the novelty and contribution of our study are not elaborated in the original manuscript. The use of earthquake scenarios for hazard analysis, and the use of physics-based numerical simulations to predict ground motions have been studied for decades. However, this study provides a fully physics-based approach for assessing the MCE hazard by comprehensively considering the source process, propagation path, and local site conditions. To the best of the authors' knowledge, it achieves two important breakthroughs over existing research in these areas based on the latest advances in deterministic broadband ground-motion simulations and our newly proposed multidimension source model for describing the rupture process.

First, this study realized the deterministic broadband analysis for the MCE hazard. We applied the newly-developed broadband ground-motion simulation method to the seismic hazard analysis procedure, driving an order increase of effective frequency contents for the deterministic ground-motion simulations, i.e., from the low frequency range (< 1 Hz) to the high frequency range up to

10 Hz, which satisfies most requirements of engineering structures. The whole physical process of earthquakes, e.g., the source process, propagation path, and local site conditions, is deterministically considered to obtain realistic ground motions at the target site.

Second, the proposed method achieved the full-scenario analysis for the MCE hazard by taking advantages of self-similar sub-sources and adjoint simulations. Previous scenario-based seismic hazard analysis methods used a very limited number of earthquake scenarios, which includes only partial source variables, and thus could not comprehensively assess the risk of possible future earthquakes. In contrast, we obtained the MCE hazard through massive earthquake scenarios (up to tens of millions of scenarios) with a holistic consideration of the uncertainties in all kinematic parameters of the source process, enabling more comprehensive and accurate prediction for the seismic hazard.

Moreover, the proposed method can generate the spatial ground-motion field for specified hazard levels around the target site by forward simulations. This is of great importance for major projects with large-span structures (e.g., dams, bridges, and nuclear power stations), as the spatial heterogeneity of ground motions may have profound influences on those structures. Furthermore, this study also revealed some essential findings through the quantitative analysis of the seismic hazards. For instance, the seismic hazards for the same site can have significant differences when they are represented by different IMs.

In a word, we believe that these contributions, along with the detailed analysis and results presented in the manuscript, make our study valuable and relevant to the field of seismic hazard analysis.

As broadband ground motions at the target site are explicitly generated for all earthquake scenarios, the seismic hazards of arbitrary IMs can be obtained, not just spectral accelerations. Therefore, for specific major engineering structures, the proposed method can provide personalized seismic hazard results by using the optimal IMs of them. Moreover, this method can give the earthquake scenarios with the specified seismic hazard levels and the corresponding broadband ground motions, which can be directly used in performance-based seismic design and analysis. The spatial ground motion field around the target site can also be generated using the earthquake scenarios, which is particularly important in the seismic design for large-span structures such as dams and bridges.

Meanwhile, the MCE hazards for the Xiluodu dam site are further analyzed with four representative IMs (Fig. S4 and S5), including the root-mean-square of acceleration (a_{RMS}),

root-mean-square of velocity (v_{RMS}), Arias intensity (I_a), and cumulative absolute velocity (CAV). The four IMs at the river-parallel direction for the Yaziba seismic zone are obviously higher than those for the Leibo seismic zone, indicating that the seismic hazards represented by different IMs may have significant differences for the same site. Therefore, with the optimal IM of specific engineering structures, this study can provide personalized seismic hazards for them, enabling more accurate and reasonable evaluations for the MCE hazards.

Moreover, using the source process of earthquake scenarios corresponding to the specified hazard level, e.g., those shown in Fig. 3, the spatial ground-motion field around the target site can be obtained by forward simulations. ShakeMaps for the spatial ground-motion field generated by forward simulations with the two earthquake scenarios in Fig. 3 are shown in Figs. S6 and S7, respectively. From these ShakeMaps, ground motions generated by realistic earthquake scenarios can show significant spatial heterogeneity, which may have profound effects on major projects with large-span structures, e.g., dams, bridges, and nuclear power stations. Overall, this study demonstrates that the physical backgrounds of earthquakes, including the source process and the propagation path, have significant impacts on the ground motions at specific sites and should be paid special attention in the seismic design and analysis there.

Comment 1.2:

"Maximum credible earthquake" is a discouraged terminology in recent decades. It was previously used to refer to earthquake ground motions before its phasing out, and it is unclear here whether the term is being used to refer to a ground motion or an earthquake magnitude (the full set of earthquake source parameters is a vector and so doesn't have a clear maximum). There is no "maximum" earthquake ground motion either, as the authors acknowledge through their use of millions of stochastic simulations. An alternate terminology would be preferred.

Response: We agree that the term of "maximum credible earthquake" in the original manuscript is not clear due to its ambiguity. We use this term to refer to the earthquake with the largest magnitude that can reasonably be expected to be generated by a specific source on the basis of seismological and geological evidence (Earthquake Design and Evaluation for Civil Works Projects, ER 1110-2-1806, 2016). We also appreciate the comment on the lack of a clear "maximum" earthquake ground motion. In this study, we used numerous earthquake scenarios to account for the uncertainties of all the kinematic source parameters. The hazards of the earthquakes with the largest magnitude, i.e., the maximum credible earthquake hazard (MCE hazards), is then obtained using broadband ground

motions generated these earthquake scenarios. With the MCE hazards, ground motions corresponding to the specified hazard levels can be directly used in performance-based seismic design and analysis. The meanings of "Maximum credible earthquake" and "Maximum credible earthquake hazards" are clarified in the revised manuscript.

Herein, the term of MCE refers to the earthquake with the largest magnitude that can reasonably be expected to be generated by a specific source on the basis of seismological and geological evidence¹⁸. Since there may be several significant active faults with different upper magnitude limits in the vicinity, the target site may have multiple MCEs. The MCE hazards, i.e., the hazards for the target site caused by the largest magnitude earthquakes at all considered active faults, are analyzed using comprehensive MCE scenarios at those active faults.

Comment 1.3:

Why generate millions of ground motion simulations, but consider only a single fixed earthquake magnitude, dip, etc. for each fault? Some thought and discussion is needed regarding uncertainties in potential future earthquake scenarios, and how they are reflected in the analysis.

Response: In this study, uncertainties of all kinematic parameters of the source process, including the fault location, the hypocenter, and the distribution of slip, rupture time, rise time, and rake, were comprehensively taken into account to obtain the MCE hazard for the target site. Moreover, we considered the earthquake scenarios with the maximum credible magnitude that is determined based on the historical earthquake records and the fault exploration results¹. The dip of the seismic fault where the earthquake scenarios occur is determined by the fault exploration results¹. However, it should be noted that our proposed method is not limited to a single magnitude and fault tendency. If needed, the uncertainty of these two factors can also be considered in the seismic hazard analysis procedure when the dip angle of fault and the maximum credible magnitude are not clear.

Comment 1.4:

Line 108, 2000 m/s at the surface is very stiff. Is this representing a measured value at the site, or is this high value necessary due to numerical modeling needs? A clarification is needed.

Response: Velocity at the dam site is determined based on the exploration results of the engineering site¹. In this study, the Xiluodu arch dam was selected as the research object. Due to the special characteristics of the arch dam, the foundation conditions are extremely demanding, and thus the engineering site of the Xiluodu arch dam is chosen on very hard bedrock. As the reviewer suggested,

the local site conditions of the Xiluodu dam are clarified in Supplementary Information of the revised manuscript.

Here it should be noted that the foundation conditions are very demanding for the arch dams due to their structural characteristics. Therefore, the engineering site of the Xiluodu arch dam is chosen on the hard bedrock where the minimum shear wave velocity is about 2000 m/s.

Reviewer #2 (Remarks to the Author):

Comment 2.1:

NOTEWORTHY RESULTS

This paper provides a case study of potential ground shaking from earthquakes on two faults close to a major dam in China, demonstrating that the hazard is similar from both faults despite the variations in potential maximum credible earthquakes. In most respects the method here is more noteworthy than the specific results for the case study, using an extremely large number of ‘scenarios’ to better define uncertainty in ground shaking at a specific site.

SIGNIFICANCE OF THE WORK

The field of earthquake hazard and ground motion simulation is large, and so such a study will no doubt be of interest to a large number of specialists. However, I’m unclear of the wider interest or implications for this work beyond providing slight decreases in output variability from existing methods. Perhaps I missed this in the Introduction section, but it’s unclear to me how the suggested approach is driving the science forward beyond incremental gains, and the focus on a very specific case study doesn’t help in this regard.

It appears to me that the main significance of the work is the ability to match hazard estimates from existing Ground Motion Prediction Equations albeit with reduced variability/uncertain. While I don’t question this is an important step forward, I’m unconvinced this is a substantially significant contribution for a multi-disciplinary audience.

Response: We thank the reviewer for the constructive comments on our manuscript. Regarding the reviewer’s concern about the wider interest and implications of our study beyond providing slight decreases in output variability from existing methods, we would like to emphasize that the proposed approach is driving the science forward beyond incremental gains. We propose a seismic hazard analysis method based on the full-scenario deterministic analysis of MCE. Compared to traditional seismic hazard analysis methods, our method can provide the hazard results for any intensity measure, not just the spectral accelerations. Therefore, for major engineering structures such as

dams, long-span bridges, and high-rise buildings, this method can give personalized seismic hazard results by using optimal intensity measures instead of the spectral accelerations that are commonly adopted in the conventional approaches.

Additionally, the proposed method can provide earthquake scenarios and broadband seismic motions corresponding to any hazard level, which can be directly used for structural response analysis. The spatial ground motion field around the target site can also be generated using the earthquake scenarios of specific seismic hazard levels, which is particularly important for large-span structures such as dams and bridges.

Moreover, our method can generate a massive amount of near-field broadband ground motions, which can be used to improve the results of empirical ground motion prediction equations for near-field large earthquakes. This is of great significance for seismic hazard analysis research because the empirical ground motion prediction equations are limited by the lack of records of the near-field large earthquake and thus their predictions for the near-field hazard of large earthquake are usually not reliable.

We acknowledge this comment regarding the focus on a specific case study, and we have added additional discussions in the manuscript to highlight the general applicability of the proposed approach beyond the specific case study presented in this work.

Compared to traditional seismic hazard analysis methods, our method can provide hazard results for any intensity measures (IMs), not just the spectral accelerations. Therefore, for major engineering structures such as dams, nuclear power plants, long-span bridges, and high-rise buildings, this method can give personalized seismic hazard results by using optimal IMs to provide more accurate seismic hazard evaluations. Herein, four representative IMs (root-mean-square of acceleration (a_{RMS}), root-mean-square of velocity (v_{RMS}), Arias intensity (Ia), and cumulative absolute velocity (CAV)) are adopted to analyze the MCE hazard. These IMs are defined as follows,

$$a_{RMS} = \left(\frac{1}{t_D} \int_0^{t_D} a(t)^2 dt \right)^{1/2} \quad (S1)$$

where t_D is the total duration of ground motions, and $a(t)$ is the acceleration time series.

$$v_{RMS} = \left(\frac{1}{t_D} \int_0^{t_D} v(t)^2 dt \right)^{1/2} \quad (S2)$$

where $v(t)$ is the velocity time series.

$$CAV = \int_0^{t_D} |a(t)| dt \quad (S3)$$

$$I_a = \frac{\pi}{2g} \int_0^{t_D} (a(t))^2 dt \quad (S4)$$

Seismic hazard results for the four representative IMs at the Xiluodu dam site for scenario earthquakes occurred in the Yazhiba seismic zone and the Leibo seismic zone are presented in Fig. S4 and S5, respectively. In contrast to the seismic hazard results represented by the spectral accelerations (Fig. 2), which are comparable for the two seismic zones, the four IMs at the river-parallel direction are obviously higher for the Yazhiba seismic zone than those for the Leibo seismic zone. It is demonstrated that the seismic hazard results for different IMs may have significant differences for the same site. Therefore, adopting the optimal IM for different engineering structures can provide more reasonable seismic hazard evaluations.

Fig. S4 Seismic hazard results for four different IMs at the Xiluodu dam site for scenario earthquakes occurred in the Yazhiba seismic zone. The four IMs corresponding to all MCE scenarios are summarized to obtain the seismic hazard results at the Xiluodu dam site. The seismic hazard results in the river-parallel direction, river-perpendicular direction, and

vertical direction are presented for the Yaziba seismic zone.

Fig. S5 Seismic hazard results for four different IMs at the Xiluodu dam site for scenario earthquakes occurred in the Leibo seismic zone. The four IMs corresponding to all MCE scenarios are summarized to obtain the seismic hazard results at the Xiluodu dam site. The seismic hazard results in the river-parallel direction, river-perpendicular direction, and vertical direction are presented for the Leibo seismic zone.

ShakeMaps generated by forward simulations for the scenario earthquakes at the Yaziba seismic zone and Leibo seismic zone, as shown in Figure 3, are presented in Fig. S6 and S7, respectively. These ShakeMaps demonstrate that the realistic spatial ground motion field around the target site can be thoroughly produced by forward simulations using the specified rupture process of earthquake scenarios corresponding to specific hazard levels. Therefore, for large-span structures (e.g., dams and bridges) that may be significantly affected by the heterogeneous ground-motion field, our method can provide more realistic ground motion input for the seismic design of these structures.

Fig. S6 ShakeMap for the scenario earthquake at the Yaziba seismic zone shown in Figure 3. PGA of the model domain for this scenario earthquake are presented by different colors. The black triangle represents the dam site.

Fig. S7 ShakeMap for the scenario earthquake at the Leibo seismic zone shown in Figure 3. PGA of the model domain for this scenario earthquake are presented by different colors. The black triangle represents the dam site.

Comment 2.2:

DOES THE WORK SUPPORT THE CONCLUSIONS AND CLAIMS

Partly. The method is able to produce a large number of simulations and produce results that are in keeping with previous findings with reduced variability. However, I'm not sure the approach makes the advances that the authors claim in the Introduction section

Response: We agree that the application scope of the proposed method is not fully elucidated in the original manuscript. Here we clarified that the comparison with the empirical GMPEs is only to verify the proposed method. Our method can provide much richer results over the conventional approach, e.g., arbitrary intensity measures and the spatial ground motion field for the specified hazard levels (please see the response to Comment 2.1), which cannot be provided by the conventional approach. Moreover, the advance of the proposed method is further elaborated in the response to Comment 1.1.

Comment 2.3:

ARE THERE ANY FLAWS IN THE DATA ANALYSIS, INTERPRETATION AND CONCLUSIONS?

In the main, this study appears robust, with suitable data analysis and interpretation. Where I am left confused is in the conclusions and how other researchers can and should take this work forward. If this is some major leap forward in the science of earthquake hazard derivation then I must be missing it, and I would recommend the authors make this point more clearly. Perhaps I am underplaying the technical advancements that have gone into achieving these results, but to me this seems like a fairly standard Monte Carlo model using existing assumptions and understanding, rather than generating some new understanding. Even for this particular case study, the results appear to simply confirm the existing knowledge from empirical GMPEs. I would encourage the authors here to make a better 'sales pitch' as to why this approach is so important, and in particular focus on making that clear and obvious to a broader audience – at present I imagine much of the detail is really only relevant for a sub-set of specialists in ground motion modelling.

Response: We appreciate the reviewer's positive feedback on the robustness of our study and our data analysis and interpretation. We are also grateful for the valuable comments on the conclusions and the importance of our work for a broader audience, and the general applicability and novelty of our proposed method are elaborated in the response to Comment 1.1 and 2.1.

Comment 2.4:

IS THE METHODOLOGY SOUND? IS THERE ENOUGH DETAIL PROVIDED IN THE METHODS FOR THE WORK TO BE REPRODUCED?

From what I can tell, the method is appropriate and robust, and the paper describes the approach in sufficient detail to allow reproduction. However, the precise details of the approach are slightly beyond my specific expertise and so other referees may have more pertinent comments.

My main concerns here are around the wider applicability and assumptions that go into the model. If the authors envisage this approach being widely adopted, then I would suggest they focus on describing how this method can be applied elsewhere, particularly in regions where there is little information on historical earthquakes and fault structures (i.e. most of the world's fault-lines!). Further, there appears to be no real scrutiny of the relationships linking basic seismic information to the MCE. So, in reality, is the main contribution of this paper the ability to provide a very large number of models using existing understanding and assumptions?

Response: We appreciate the reviewer's positive feedback on the robustness and reproducibility of this study. The meaning of MCE in this study and the main contributions of our proposed method have been described in the response to Comment 1.2. Herein, we emphasize that our proposed

method has good universality and is applicable to different regions. Similar to the conventional approach, our method can be applied to analyze the seismic hazard at the target sites with some basic seismic information in the region, e.g., historical earthquakes and fault structures. Nevertheless, we believe that this does not overly restrict the proposed method because, these basic seismic information is usually available for densely populated urban areas and major engineering sites that are the focus of seismic hazard analysis.

Reviewer #3 (Remarks to the Author):

Comment 3.1:

The paper is interesting and well written and covers an important topic. It deals with the estimation of the MCE at a dam site using physics-based ground motion.

Description

The authors state that the maximum frequency of the simulations is up to 10Hz and that all important variability of the parameters influencing the results is considered. They selected the faults and assigned the maximum expected magnitude. Running thousands of scenarios by changing source parameters, they computed accelerograms at the edge of a site by extracting relevant intensity measurements. They evaluated the distribution of the different MIs. They claim that the results express the MCE.

Observations:

The topic is quite interesting as both MCE assessment and the use of physics-based ground motion simulations are active research topics. However, more details about specific input patterns need to be added. Also, authors must state what they mean by MCE. Since it is a seismic input for the structural analysis, it should represent the input that provides the maximum excitation for the structure under analysis. Since the dam is very stiff, it is possible that scenarios with lower magnitudes could excite the dam more than the scenarios considered here. So is it really MCE?

To really find the MCE as an input for the structural analysis, multiple scenarios should be combined, also considering lower magnitudes. A procedure to account for this, also through ground motion simulations, can be found in the reference attached to the end.

Response: We are grateful for the reviewer's suggestion regarding MCE. As mentioned in the response to Comment 1.2, MCE refers to the earthquake with the largest magnitude that can reasonably be expected to be generated by a specific source on the basis of seismological and geological evidence (Earthquake Design and Evaluation for Civil Works Projects, ER 1110-2-1806, 2016). Meanwhile, we have also provided additional details about the specific input patterns.

We also fully agree with the reviewer that multiple earthquake scenarios with different magnitudes should be considered to really find the MCE for the target site. According to the seismic safety evaluation report of engineering site for the Xiluodu arch dam ¹, there are two important fault structures in the near field: the Yaziba fault with a potential magnitude limit of 7.5 and the Leibo fault with a potential magnitude limit of 7.0. Earthquake scenarios occurred at the two fault structures are considered to analyze the MCE hazard for the Xiluodu dam site. We found that although the magnitude of scenario earthquakes at the Leibo fault ($M_w7.0$) is significantly less than that of scenario earthquakes at the Yaziba fault ($M_w7.5$), scenario earthquakes at the Leibo fault can cause similar seismic risk to the Xiluodu dam site. As the reviewer suggested, following ², we further combined the ground motions generated by the scenarios at the two considered fault structures to find the final MCE for the target site. The seismic hazards for the Yaziba seismic zone, the Leibo seismic zone, and the combination of the two seismic zones are shown in Fig. R1. However, the seismic hazards of different faults may differ significantly when different IMs are adopted (the response to Comment 2.1). Considering that the structural dynamic response analysis requires the ground-motion time series as inputs, the seismic hazards represented by just a single IM may not be strongly associated with the real dynamic response of the structures, because the ground-motion time series with the same IM but different physical backgrounds can result in different structural responses³. Therefore, we think it may be more reasonable to evaluate the seismic hazards of different faults separately and select the ground-motion time series corresponding to the specified hazard levels at these faults for the structural dynamic analysis.

Fig. R1 Seismic hazards of the Xiluodu dam site. Spectral accelerations of all MCE scenarios are summarized to obtain the seismic hazard results at the specified periods. The seismic hazards in the river-parallel direction and the river-perpendicular direction are presented for the Yaziba seismic zone, the Leibo seismic zone, and the combination of the two seismic zones.

Comment 3.2:

The uncertainty in the structural model has not been considered (and this may explain part of the reduced variability with respect to the GMPE) and the distribution of the parameters implemented in the source model is not well described.

In short, the paper presents interesting aspects but lacks a thorough discussion on the possible limitations of the presented results. A minor revision is therefore suggested to clarify the following specific comments.

Response: We agree with the reviewer that the epistemic uncertainty in the structural model can contribute to the variability of the seismic hazard results. However, with further refinement of the velocity models, these differences are expected to diminish ⁴. In our previous studies of ground-motion simulations for the 1992 Landers earthquake and the 1994 Northridge earthquake ^{5,6}, we found that by coupling the regional velocity model with the local geophysical information, the simulated ground motions achieve a fairly good agreement with the recorded ground motion in a wide frequency range (0-8 Hz for the 1992 Landers earthquake and 0-5 Hz the 1994 Northridge earthquake). Therefore, we did not consider the epistemic uncertainty of the structural model in this study. Meanwhile, we added a detailed description of the parameters implemented in the source model in the revised manuscript.

Specific comments:

Comment 3.3:

- The structural model is not described

Response: The structural model is further described in the supplementary materials of the revised manuscript.

To simulate the seismic wave propagation of source-path-site, a three-dimensional (3D) numerical model for the vicinity of the Xiluodu dam site is constructed with the high-resolution topography and realistic velocity structures. The topography is considered using the digital elevation model data with a resolution of 30 m. Since this study only focuses on ground motions at the engineering site, accurate local velocity structures are of great importance for generating realistic ground motions there¹. In this study, we construct a three-dimensional (3D) numerical model (Fig. S1) by coupling the regional velocity model, East Asia EARA2014, with the geological exploration information at the dam site. Here it should be clarified that the foundation conditions are very demanding for the arch dams due to their structural characteristics. Therefore, the engineering site of the Xiluodu arch dam is chosen on the hard bedrock with the minimum shear wave velocity of about 2000 m/s.

Fig. S1 Velocity structures around the Xiluodu dam site. The shear wave velocity in the model domain is presented by different colors as shown in the color bar. The black triangle indicates the Xiluodu dam site.

Comment 3.4:

- The velocity seems to be too fast to reach the specific conditions of the real site, please look into this

- The maximum frequency is a function of the source, the structural model and the precision of the site specific layers. The high frequencies are strongly influenced by the local soft layers. It seems they are not included in the model

Response: Velocity at the site is determined based on the geological exploration data of the engineering site. The Xiluodu dam site is chosen on very hard bedrock. According to the geological exploration ¹, there are no soft soil layers at the dam site. We have found that the simulated ground motions can achieve a fairly good agreement with the records by coupling the regional velocity model with the local geophysical information ^{5,6}. As the Xiluodu dam site has no local soft layers, the nonlinear material behaviors there will be slight, and the obtained ground motions in this study should be relatively reasonable.

Comment 3.5:

- Line 129: An occurrence model for magnitudes has not been used so it is better not to call the IM distributions "hazard curves".

Response: As the reviewer suggested, the IM distribution is renamed “seismic hazards”.

Comment 3.6:

- Line 155: it is true that accelerograms are available, however since it is not imaginable to use all of them, it is necessary to apply an accurate selection procedure based on the dynamic properties of the structure to be analysed.

Response: We are grateful for this insightful and valuable suggestion on applying our method to the practice of structural dynamic analysis. Our method generates numerous ground-motion time series (up to tens of millions in this study), but the structural dynamic analysis can only use a very limit number of them. We fully agree with the reviewer that a selection method is needed to apply the generated ground-motion time series to the structural dynamic analysis. Currently, we are studying the application of this method to the practice of the performance-based seismic design and analysis, e.g., generating the physics-based conditional mean spectrum and the associated ground-motion time series. However, as this study mainly aims to develop a seismic hazard analysis method for MCE and limited by the space, we will investigate the selection method for ground-motion time series in a subsequent study.

Comment 3.7:

- Fig. 4: it is true that the PBGMS could be a tool to obtain more information on seismic hazard and also to limit uncertainties. But in this case the variability appears to be very small. A discussion of this needs to be added. Uncertainty on soil parameters has also not been included and this would surely increase dispersion (particularly at high frequency)

Response: Parameters of the local site are described in the response to Comment 1.4, and the variability of the seismic hazards is discussed in the response to Comment 3.2.

Comment 3.8:

- Line 210. See previous discussion on “What is MCE?”

Response: In this study, MCE refers to the earthquake with the largest magnitude that can reasonably be expected to be generated by a specific source on the basis of seismological and geological evidence. It is now added in the revised manuscript.

Comment 3.9:

- Line 217: At least in the supplementary material it might be helpful to explain the parameter distributions and how that variability was evaluated.

Response: As the reviewer suggested, the details about parameter distributions and variability are provided in the supplementary materials.

Table S2 Distribution range of the source process parameters for earthquake scenarios at the Yaziba fault and the Leibo fault.

Fault name	Parameter	Distribution range	Constraint
Yaziba	Hypocenter	All locations on the fault	—
	Slip (m)	0-6.72	Average=1.68
	Rupture velocity (km/s)	2.5-3.5	—
	Rise time (s)	1-4	Average=2.5
	Rake (deg)	40-140	70≤Average≤110
Leibo	Hypocenter	All locations on the fault	—
	Slip (m)	0-3.0	Average=0.75
	Rupture velocity (km/s)	2.5-3.5	—
	Rise time (s)	0.5-2.5	Average=1.5
	Rake (deg)	130-230	160≤Average≤200

Table S2 presents the distribution range of the source process parameters for earthquake scenarios at the Yaziba fault and the Leibo fault. For the generated earthquake scenarios, the hypocenters of them can be anywhere on the fault ruptures of earthquake scenarios. Based on the constraints on the slip, rupture velocity, rise time, and rake angle ranges in the source inversion study⁶ and the stochastic source process model⁷, the source parameter ranges of the earthquake scenarios are defined in this study. The slip amount of each subfault is allowed to vary from 0 to 4 times the average slip value for the earthquake scenarios, while the average of them remains constant, i.e., 1.68 m for the Yaziba fault and 0.75 m for the Leibo fault. By solving the Eikonal equation, the rupture time of each subfault is derived from the distribution of rupture velocity that varies from 2.5 km/s to 3.5 km/s. The rise time of each subfault is allowed to vary from 1 s to 4 s for the Yaziba fault and 0.5 s to 2.5 s for the Leibo fault, while the average of them is equal to 2.5 s for the Yaziba fault and 1.5 s for the Leibo fault, respectively. According to the fault exploration results, the Yaziba fault is a reverse fault while

the Leibo fault is a strike-slip fault. To determine the rake, the average rake is firstly allowed to vary from 70° to 110° for the Yaziba fault and from 160° to 200° for the Leibo fault. Next the rake of each subfault is allowed to vary in the range of $\pm 30^\circ$ around the average rake.

Reference:

1. Seismic safety evaluation report for the project site of Jinsha River Xiluodu Hydropower Station (in Chinese). Beijing Zhongzhen Pioneering Engineering Technology Research Institute, (2011).
2. Hassan, H. M. et al. Site-specific ground motion modeling for a historical Cairo site as a step towards computation of seismic input at cultural heritage sites. *Engineering Geology* **268**, 105524 (2020).
3. Wang, J.-T., Jin, A.-Y., Du, X.-L. & Wu, M.-X. Scatter of dynamic response and damage of an arch dam subjected to artificial earthquake accelerograms. *Soil Dynamics and Earthquake Engineering* **87**, 93–100 (2016).
4. Baker, J. W., Luco, N., Abrahamson, N. A., Maechling, P. J. & Olsen, K. B. ENGINEERING USES OF PHYSICS-BASED GROUND MOTION SIMULATIONS. in *Proceedings of the 10th National Conference in Earthquake Engineering* (2014).
5. Wang, X.-C., Wang, J.-T., Zhang, L. & He, C.-H. Broadband ground-motion simulations by coupling regional velocity structures with the geophysical information of specific sites. *Soil Dynamics and Earthquake Engineering* **145**, 106695 (2021).
6. Wang, X., Wang, J. & Zhang, C. A Broadband Kinematic Source Inversion Method Considering Realistic Earth Models and Its Application to the 1992 Landers Earthquake. *JGR Solid Earth* **127**, (2022).

REVIEWER COMMENTS

Reviewer #1 (Remarks to the Author):

I am still not convinced that this manuscript makes the type of advancement that is typical of a Nature Communications paper. Using numerical simulations to characterize ground motion hazard is an active field with many similar studies already in existence, and choice here of what uncertainties to consider and what to omit is seemingly arbitrary.

Reviewer #2 (Remarks to the Author):

The authors have done a commendable job here addressing each of the reviewers comments, and this current version of the manuscript is significantly improved compared to the original. I'm pleased to see the authors have now more clearly described the wider contribution this paper intends to make to the field, with a reduced focus on the specific case study, which is now appropriately highlighted as an exemplar of the approach.

There are still some minor areas that could be addressed to improve the paper and its readability.

First, the term Maximum Credible Earthquake still seems fraught to me. The other reviewers highlighted this initially and the authors have responded and now clearly define what they mean by MCE. However, given the context of the paper is really on the ground shaking at any given site, I find the focus on a single magnitude confusing. The authors themselves note that even moderate earthquakes have caused substantial impacts in the past. I remain unconvinced that we should simply take at face value that the biggest magnitude earthquake can cause the largest shaking at a given site, considering complex interactions of seismic waves with ground properties. My understanding is that this is actually quite a minor point and that the proposed method could easily consider different magnitudes for each fault, but the authors here simply choose to demonstrate their approach via the "MCE". My issue therefore is that by choosing to focus on the term MCE the authors detract from their own work via readers focussing too intently on this term, rather than engaging fully with the method proposed.

Secondly, in the supplementary information (and in the response to reviewers) the authors show figures of ground motion from one of their scenario earthquakes, but include no information on what parameters were involved in this scenario. In order for these figures to make sense, its important that readers are provided those key details - which of your 22 million scenarios is this? In addition, I wonder if the authors have considered a figure (or figures) here showing the spatial variation in ground shaking across all their scenarios? I think this would be a valid contribution.

Finally, I am surprised to see so many of the pertinent methods details buried in the supplementary information and not the main text. I appreciate the word limit and requirements of Nat Comms, but to

me some of the most important details and descriptions are buried away here when they really belong in the main text. The authors have done a solid job of addressing the reviewers comments through changes to the SI, but I think the paper would benefit from including parts of this in the main manuscript itself.

Overall, these are relatively minor and I believe the authors have done a reasonable job of addressing the concerns previously posed.

Reviewer #4 (Remarks to the Author):

The manuscript presents a comprehensive understanding of the significant threat posed by major earthquakes to modern society, emphasizing their destructive potential and inherent unpredictability. The concept of the maximum credible earthquake (MCE) for a specific fault, characterized by various potential scenarios with differing source processes, contributes to the complex and uncertain nature of future seismic hazards. The abstract introduces a solution in the form of a full-scenario analysis method that employs deterministic broadband simulations across a multitude of scenarios. This approach seeks to address the intricate nature of MCE hazards by considering all possible source process parameters along the fault, thereby facilitating a more comprehensive assessment.

While the abstract successfully explains this novel full-scenario analysis method and applies it to the seismic hazard assessment of China's Xiluodu dam, there appears to be potential for improvement in terms of practical application. Furthermore, the proposed method's capacity to give a variety of intensity measures, ground motion time series, and spatial ground-motion fields hold promise for a more precise and realistic evaluation of MCE hazards, with possible applications in earthquake engineering.

In conclusion, the paper presents an intriguing approach to tackling the uncertainty associated with MCE hazards and it would be appropriate to accept it for publication. Although the method's conceptual foundation and potential contributions are commendable, a closer examination and improvement in the application phase could elevate the paper's overall impact.

1. How does the incorporation of numerous potential scenarios with different source processes contribute to the uncertainty in predicting maximum credible earthquakes?
2. Can you elaborate on the significance of conducting deterministic broadband simulations as part of the proposed full-scenario analysis method?
3. Could you provide more insight into the specific parameters and variables considered within the full-scenario analysis approach for evaluating MCE hazards?
4. In what ways could the refinement of the practical application aspect enhance the practicality and adaptability of the proposed method in real-world scenarios?
5. Considering the potential for more realistic MCE hazard evaluations, how might the proposed method influence decision-making processes in earthquake engineering projects, particularly in areas prone to seismic activity?

Response to the comments

The authors would like to thank the reviewers for their constructive comments, which were of great help in improving the manuscript. The authors have carefully considered the detailed comments, and the point-by-point responses to the reviewers' comments are given below (Reviewers' comments are in *italic fonts*, responses are in normal fonts and changes of the manuscript are in **bold fonts**).

REVIEWER COMMENTS

Reviewer #1 (Remarks to the Author):

Comment 1.1

I am still not convinced that this manuscript makes the type of advancement that is typical of a Nature Communications paper. Using numerical simulations to characterize ground motion hazard is an active field with many similar studies already in existence, and choice here of what uncertainties to consider and what to omit is seemingly arbitrary.

Response: We sincerely appreciate the reviewer for providing the valuable comments on our work. The comments in this and the original review have indeed guided us in refining and enhancing the manuscript. We regret that the previous revision may not have adequately highlighted the contributions and advancements of our study. Herein, we would like to further elaborate the advancements and contributions of this study over existing research on evaluating the seismic hazard using earthquake scenarios and numerical simulations. Moreover, the uncertainties considered in this study are also interpreted below.

(a) Advancements and contributions of this study

We acknowledge that there have been many studies that use earthquake scenarios and numerical simulations to predict the seismic hazard (Graves et al., 2011; Denolle et al., 2014; Robinson et al., 2018; Panza and Bela, 2020; Milner et al., 2021; Zhang et al., 2021). As Baker et al. (2014) pointed out, “numerical simulations of earthquake strong ground motions have improved to the point where it is worth investigating the predictive power of these physics-based methods in seismic hazard analysis”. However, there are still some crucial issues that limit their applications in real-world scenarios, such as the restriction to low-frequency range (Baker et al., 2014). In this study, we address two key issues constraining the applications of deterministic ground-motion simulations in seismic hazard evaluations.

First, current seismic hazard evaluations using earthquake scenarios and numerical simulations are mostly constrained to the low-frequency range. For instance, 0-0.5 Hz in (Graves et al., 2011; Milner et al., 2021), 0-1 Hz (Panza and Bela, 2020; Zhang et al., 2021). However, high-frequency

ground motions are of most interest for earthquake engineering practice (Fu et al., 2017; Baker et al., 2021). Although some efforts have been made for fully deterministic broadband ground-motion simulations, e.g., 0-4 Hz for a sequence of small earthquakes ($M_w < 4.4$) in the Volvi basin of Greece (Maufroy et al., 2015), 0-4 Hz for a $M 7.0$ scenario earthquake (Rodgers et al., 2018), 0-5 Hz for 2016 $M_w 6.5$ Norcia, Italy, Earthquake (Pitarka et al., 2022), these methods are inhibited by the complexity of implementation and substantial computational efforts. Consequently, it is difficult to effectively account for the inherent uncertainties of earthquakes using these broadband simulation methods, and this limits the usability of these methods in applications such as seismic hazard analysis and source inversion (Wang et al., 2022).

Second, to describe the complex source process of earthquakes, especially that is associated with high-frequency seismic waves, the source models in deterministic ground-motion simulations need to include fine enough scales, leading to a large number of variables, which can be up to tens of thousands or even hundreds of thousands for moderate or large earthquakes (Graves and Pitarka, 2010; Anderson, 2015; Crempien and Archuleta, 2015). It is recognized that the ensemble of earthquake scenarios should be large enough to comprehensively capture the variability in the source parameters of earthquakes (Baker et al., 2014). However, existing studies only consider partial source variables, e.g., hypocenter and slip distribution (Graves et al., 2011), and use a very limited number of earthquake scenarios to represent the potential future earthquakes, such as, less than 100 scenarios (Denolle et al., 2014; Robinson et al., 2018), a total of 415000 scenarios for 7000 ruptures (that is, less than 60 scenarios for each rupture) (Graves et al., 2011). With such a number of scenarios and only partial variables considered, it is challenging to comprehensively evaluate the seismic hazard of potential future earthquakes, which has a large number of rupture variables (even up to hundreds of thousands for simulating broadband ground motions).

In this study, we propose a ground-breaking way for the seismic hazard analysis based on earthquake scenarios and numerical simulations. It is conducted based on our newly developed deterministic broadband ground-motion simulation method (Wang, Wang, Zhang, and He, 2021; Wang, Wang, Zhang, Li, et al., 2021; Wang et al., 2022), which has been validated through comparisons with historical earthquake records and empirical ground-motion prediction models. Particularly, with the inverted rupture process for the 1992 Landers earthquake, the simulated ground motions of our broadband numerical simulation method even had an impressive agreement in waveforms with the records in a wide range of frequency, as shown in Figure R1. Compared to existing studies, our method achieves significant advancements in the following ways:

- By employing the developed ground-motion simulation method, we have achieved significant breakthroughs in seismic hazard evaluations by expanding the effective frequency contents

from the low frequency range (< 1 Hz) to the high frequency range up to 10 Hz. This broad frequency range meets the current requirements of seismic design for most engineering structures.

- To the best of our knowledge, this study has achieved, for the first time, a deterministic full-scenario analysis for the seismic hazard by applying the self-similar feature of our proposed multidimension source model. The seismic hazard is evaluated by considering all uncertainties of potential future earthquakes with sufficient scenarios ((up to tens of millions of scenarios or even more), enabling more comprehensive and accurate prediction for the seismic hazard.
- Our method provides much richer outputs compared to traditional seismic hazard analysis methods. For instance, it can generate a wide range of intensity measures, ground-motion time series, and spatial ground-motion fields with specific hazard levels. These enriched outputs pave the way for realistic and customized assessments of seismic hazard for specific engineering structures.

Fig. R1. Comparison of the velocity time series, Fourier velocity spectra, and acceleration response spectra for 5% damping between the records and the simulated ground motions at two representative stations in the 1992 Landers earthquake (For each station, left: E-W; right: N-S). The velocity time series of both the records and the simulated ground motions are bandpass filtered between 0.1 and 8Hz (Wang et al., 2022).

With the above-mentioned advancements, our method has great potential in real-world scenarios and may have significant implications in the field of earthquake engineering. Here we list two important potential applications of our method in the seismic design practice:

(1) Accurate evaluations of seismic hazards for major projects near significant faults. By incorporating the realistic physical process of earthquakes and Earth models, our method assesses the seismic hazard based on deterministic broadband ground-motion simulations, enabling to consider complex physical phenomena such as the rupture directivity, topographic effects, and basin effects. Specifically, our method has also been applied to evaluate the safety of several planned major projects which are located in seismically active areas in China and has revealed some significant phenomena. A typical example is provided in the response to Comment 4.6.

(2) Seismic design fully based on realistic and unified physical backgrounds. Except for providing the seismic hazard results, our method can further obtain spatial ground-motion fields through forward simulations, which is important for the seismic design of major projects with large spatial scales such as dams and bridges. Recently, we have proposed a source-to-structure simulation method by integrating regional ground-motion simulations and structural dynamic response analysis methods (Zhang et al., 2020), and used the fault-to-structure simulation to reasonably reproduce the dynamic response of the Pacoima dam during the 1994 Northridge earthquake (Zhang et al., 2023). Therefore, with the fault-to-structure simulation technique, our study is promising for more ambitious goals to achieve the seismic design fully based on realistic and unified physical backgrounds of earthquakes, from seismic hazard evaluations to structural dynamic analysis.

Based on fully deterministic broadband ground-motion simulations, to the best of our knowledge, this study for the first time achieved a deterministic full-scenario analysis of seismic hazard by considering all uncertainties of the earthquake source through an extensive range of scenarios. The proposed method is promising for more realistic and accurate evaluations of seismic hazards and has great application potentials in earthquake engineering. We believe that these advancements and contributions in our study are of great interest to the readership of Nature Communications.

(b) Interpretation on the uncertainties considered in this study

We are sorry that the choice of what uncertainties to consider and what to omit is not clearly described in the manuscript. It is recognized that the physical process of earthquakes primarily involves three parts, i.e., the source process, propagation path, and local site conditions. Among these, the propagation path and local site conditions are determined by the velocity structure and topography. For a specific site, both the velocity structure and topography are fixed, although with potential limitations in accuracy, particularly for regional velocity structure. Uncertainties of the propagation path and local site conditions are then considered as epistemic uncertainties, which are expected to be improved with further refinement of the Earth model (Baker et al., 2014). Therefore,

this study does not consider the uncertainties associated with the propagation path, and local site conditions, i.e., the epistemic uncertainties in the velocity structure and topography. Instead, we mainly focus on the uncertainties associated with the seismic source.

Given the current understanding of earthquakes, the sources of potential future earthquakes are highly uncertain with great randomness. Consequently, we evaluate the seismic hazard of the target site by considering the possible scenarios of potential future earthquakes. It should be highlighted that our method is able to take into account all variables of earthquakes, such as the fault location, hypocenter, and rupture process (the distribution of slip, rupture time, rise time and rake on the seismic fault), which are considered in the case of Xiluodu dam as presented in the manuscript. In relation to the other variables such as earthquake magnitude and fault dip, we want to provide some clarification. As highlighted by Reviewer 2 in Comment 2.2, our approach is adaptable and can readily incorporate these variables as well. For the presented case of Xiluodu dam in this study, however, we adopted the maximum magnitude of the surrounding faults for the consideration of extreme earthquakes that the site might encounter, and the fault dips are determined based on the explorations of these faults. If required, our method is flexible enough to incorporate different earthquake magnitudes and fault dips for evaluating the seismic hazard of the target site.

In summary, except for epistemic uncertainties related to Earth models, which are expected to diminish with further refinement of regional velocity structures, our method is equipped to incorporate all uncertainties related to the potential future earthquakes. This comprehensive consideration of uncertainties demonstrates the robustness and adaptability of our approach in the face of the unpredictable nature of earthquakes. We added the following text to the revised manuscript to describe the uncertainties considered in this study.

Additionally, it is worth emphasizing that our method is capable of incorporating all uncertainties related to potential future earthquakes. In this study, uncertainties of the fault location, hypocenter, and rupture process of earthquakes are considered in the seismic hazard evaluations for the Xiluodu dam site, while the earthquake magnitude are fixed to the maximum magnitude of surrounding faults for the consideration of MCEs, and fault dips are determined based on fault explorations. However, our method is flexible enough to incorporate other variables of earthquakes such as earthquake magnitude and fault dip if necessary.

Reviewer #2 (Remarks to the Author):

Comment 2.1

The authors have done a commendable job here addressing each of the reviewers comments, and this current version of the manuscript is significantly improved compared to the original. I'm pleased to see the authors have now more clearly described the wider contribution this paper intends to make to the field, with a reduced focus on the specific case study, which is now appropriately highlighted as an exemplar of the approach.

There are still some minor areas that could be addressed to improve the paper and its readability.

Response: We sincerely appreciate the reviewer for the encouraging comments and positive feedback on the improvements made in the revised manuscript. We agree that there are still some minor areas that could be further enhanced to improve the manuscript. A point-by-point response to the comments is attached below.

Comment 2.2

First, the term Maximum Credible Earthquake still seems fraught to me. The other reviewers highlighted this initially and the authors have responded and now clearly define what they mean by MCE. However, given the context of the paper is really on the ground shaking at any given site, I find the focus on a single magnitude confusing. The authors themselves note that even moderate earthquakes have caused substantial impacts in the past. I remain unconvinced that we should simply take at face value that the biggest magnitude earthquake can cause the largest shaking at a given site, considering complex interactions of seismic waves with ground properties. My understanding is that this is actually quite a minor point and that the proposed method could easily consider different magnitudes for each fault, but the authors here simply choose to demonstrate their approach via the "MCE". My issue therefore is that by choosing to focus on the term MCE the authors detract from their own work via readers focussing too intently on this term, rather than engaging fully with the method proposed.

Response: We fully agree with the reviewer that the proposed method can consider any magnitude for the given faults, not only the biggest magnitude. Our method is highly adaptable, capable of incorporating all variables of earthquakes (certainly including different earthquake magnitudes) in the evaluations of the seismic hazard of the target site. However, due to the complexity and computational cost of implementation, the proposed method is more suitable for assessing the risk of major engineering projects subject to extreme earthquakes occurring in the near field of the target site. Meanwhile, the seismic design of major engineering projects, e.g., high dams and nuclear power stations, particularly emphasizes their safety under extreme seismic conditions. Therefore, in this study, following the guidelines for determining MCE, e.g., regulation for Earthquake Design

and Evaluation for Civil Works Projects (ER 1110-2-1806, 2016) in America and code for Seismic Design of Hydraulic Structures of Hydropower Project (NB 35047, 2015) in China, we take earthquakes with the maximum magnitude at the surrounding faults as the extreme earthquakes, i.e., MCEs, to be considered. Certainly, as the reviewer pointed out, our method can easily consider earthquake scenarios with different magnitudes in the seismic hazard analysis for the target sites. In subsequent research, we plan to address the consideration of earthquakes with different magnitudes, particularly in areas where the threat of moderate to small-scale earthquakes are also seriously concerned, such as densely populated urban areas.

For the reviewer's another concern that if the biggest magnitude earthquake causes the largest shaking at a given site, we agree that magnitude is not the only determining factor for the ground-motion intensity at the specified site. As revealed in the manuscript (Fig. 2), due to the complex physical process of earthquakes from source to site, earthquakes with identical magnitude, which also occur at the same fault, can generate ground motions with considerably different intensities at the target site. However, for the same fault, under the same probability exceedance level, earthquakes with higher magnitude generally indicate more severe seismic hazards at the same site. To substantiate this conclusion, we analyzed the seismic hazard posed by earthquakes with the magnitude of $M_w7.0$, which occur at the main rupture of the Yaziba fault (F5 in the Yaziba seismic zone in Fig. S2 of the Supplementary Information), using 1 million scenarios. Fig. R2 shows comparison of the seismic hazards of earthquakes occurring at the Yaziba fault with the magnitudes of $M_w7.5$ and $M_w7.0$, respectively. It can be found that for the same exceeding probability, the PGA value of $M_w7.5$ earthquakes is obviously larger than that of $M_w7.0$ earthquakes. This phenomenon is understandable. Assume an ideal case that a smaller magnitude earthquake occurs at a given fault. Then, imagine the same fault occurs a larger magnitude earthquake with the same rupture process in the area where the smaller earthquake rupture, but with the proportionally larger slip. In this case, additional rupture and energy release occur for the larger earthquake. It is evident that the larger earthquake will surely generate stronger ground motions at the given site beyond what is observed for the smaller earthquake, due to the additional released energy. Therefore, assuming the propagation path and local site conditions remain unchanged, a larger magnitude earthquake generally has a greater threat to the target site compared to a smaller magnitude earthquake occurring at the same fault.

However, when it comes to different faults, the ground-motion intensity at a site cannot be solely determined by magnitude due to various additional factors, e.g., fault type (strike-slip or thrust), distance from the fault to the site, relative site position from the fault (hanging wall or footwall), and the medium properties along the propagation path. In particular, it is difficult to judge

whether larger magnitude earthquakes occurring farther away generate stronger ground motions at the given site compared to smaller magnitude earthquakes occurring closer to the site. Therefore, in this study, to find the most severe ground motion at a specific site, earthquake scenarios with the maximum magnitude for each significant fault in the surrounding area, i.e., the $M_w7.5$ earthquakes at the Yaziba fault and the $M_w7.0$ earthquakes at the Leibo fault, are considered separately.

Fig. R2 Comparison of the seismic hazards of earthquakes occurring at the Yaziba fault with the magnitudes of $M_w7.5$ and $M_w7.0$, respectively.

Due to the complexity and computational cost of implementation, the proposed method is more suitable for evaluating the seismic hazard of major engineering projects subject to extreme earthquakes occurring in the near field of the target site. Given that the seismic design of major engineering projects, such as high dams and nuclear power plants, particularly emphasize their safety under extreme seismic conditions, the proposed method is currently intended for evaluating the seismic hazard of major engineering projects under MCEs.

Comment 2.3

Secondly, in the supplementary information (and in the response to reviewers) the authors show figures of ground motion from one of their scenario earthquakes, but include no information on what parameters were involved in this scenario. In order for these figures to make sense, it is important that readers are provided those key details - which of your 22 million scenarios is this? In addition, I wonder if the authors have considered a figure (or figures) here showing the spatial variation in ground shaking across all their scenarios? I think this would be a valid contribution.

Response: The ShakeMaps in Figure S6 and S7 of Supplementary Information are generated by the

earthquake scenarios shown Figure 3 of the original manuscript. As the reviewer suggested in Comment 2.4, we have included these ShakeMaps in Figure S6 and S7 into the main text to afford readers a clearer understanding of our results.

In addition, we appreciate the reviewer for the constructive comment on investigating the spatial variation in ground motions across all considered scenarios, which is of great importance for evaluating the potential seismic hazard throughout the study area. However, to obtain the ground motions at the target site for all considered earthquake scenarios, it requires only 3 complete numerical simulations on the supercomputer to calculate the required strain Green's tensor from all possible source locations to the target site. In contrast, each ShakeMap of an earthquake scenario needs an expensive forward simulation on the supercomputer. Therefore, generating ShakeMaps for the 22 million earthquake scenarios considered in this study is not feasible, which requires 22 million complete numerical simulations. To investigate the spatial variation of the ground-motion field, based on the obtained seismic hazard results, we simulated 5 scenarios for each of the 3 representative seismic hazard levels (i.e., the exceedance probabilities of 10%, 50%, and 90%) for both the Yaziba fault and the Leibo fault. A total of 30 scenarios were used to generate ShakeMaps by forward simulations. Analysis of the spatial variations of the ground-motion fields is added to the Supplementary Information as follows.

In order to explore the spatial variations within the ground-motion field, 5 scenarios for each of 3 representative seismic hazard levels (i.e., exceedance probabilities of 10%, 50%, and 90%) are derived based on the obtained seismic hazard results. These scenarios occurring at the main rupture (F5 in Fig. S2) of both the Yaziba and Leibo faults are then simulated, yielding a total of 30 scenarios (15 for each fault). ShakeMaps of these scenarios are subsequently produced through forward simulations. We then compute the average (Fig. S7), standard deviation (Fig. S8), and variation coefficient (Fig. S9) for the ShakeMaps of these earthquake scenarios at both the Yaziba and Leibo faults. From Fig. S7-S9, the intensity of ground motions gradually decreases from the upper edge of the fault outward, but exhibits complex spatial distributions in the model domain. In particular, consistent with previous research⁸, ground motions can be significantly amplified at the tops of hills and ridges, while it may decrease at the valleys. Additionally, the distribution of standard deviation and variation coefficient reveals much greater variability of the ground-motion intensity in the near-fault regions for different earthquake scenarios, indicating that the rupture process of earthquake scenarios has significant effects on the ground motions in regions close to the faults. Therefore, the influence of the earthquake rupture process should be paid special attention in

the evaluations of seismic hazard for the near-fault regions.

Fig. S7 Average of the ShakeMaps of earthquake scenarios at the Yaziba fault (a) and the Leibo fault (b). The average of PGA in the model domain are presented by different colors. The red triangle represents the dam site. The top edges of the fault planes are indicated by the red solid lines, while the bottom edges of them are represented by the red dotted lines.

Fig. S8 Standard deviation of the ShakeMaps of earthquake scenarios at the Yaziba fault (a) and the Leibo fault (b). The standard deviation of PGA in the model domain are presented by different colors. The red triangle represents the dam site. The top edges of the fault planes are indicated by the red solid lines, while the bottom edges of them are represented by the red dotted lines.

Fig. S9 Variation coefficient of the ShakeMaps of earthquake scenarios at the Yaziba fault (a) and the Leibo fault (b). The Variation coefficient of PGA in the model domain are presented by different colors. The red triangle represents the dam site. The top edges of the fault planes are indicated by the red solid lines, while the bottom edges of them are represented by the red dotted lines.

Comment 2.4

Finally, I am surprised to see so many of the pertinent methods details buried in the supplementary information and not the main text. I appreciate the word limit and requirements of Nat Comms, but to me some of the most important details and descriptions are buried away here when they really belong in the main text. The authors have done a solid job of addressing the reviewers comments through changes to the SI, but I think the paper would benefit from including parts of this in the main manuscript itself.

Overall, these are relatively minor and I believe the authors have done a reasonable job of addressing the concerns previously posed.

Response: We appreciate the reviewer for this insightful comment regarding the placement of crucial details and descriptions of our method in the Supplementary Information. We agree that these important contents should be accessible within to the main text instead of the Supplementary Information. In response to this suggestion, we have revisited the manuscript and incorporated significant aspects of the proposed method and pertinent findings into the main text of the revised manuscript, including descriptions on the input parameters and the ShakeMaps of the regional ground-motion field. We believe this will enhance the clarity and overall impact of our study.

Furthermore, by employing the source process of earthquake scenarios corresponding to the specified hazard level, the spatial ground-motion field around the target site can be derived by forward simulations. ShakeMaps generated by forward simulations for the earthquake scenarios at the Yaziba fault and Leibo fault, as shown in Fig. 3, are presented in Fig. 4. From the ShakeMaps, it is demonstrated that the realistic spatial ground motion field around the target site can be thoroughly produced by forward simulations using the rupture process of earthquake scenarios corresponding to specific hazard levels. Therefore, for large-span structures (e.g., dams and bridges) that may be significantly affected by the heterogeneous ground-motion field²⁸, our method can provide more realistic ground motion input for the seismic design of these structures.

Fig. 4 ShakeMaps of earthquake scenarios at the Yaziba fault (a) and the Leibo fault (b). The ShakeMaps are generated by forward simulations using the earthquake scenarios presented in Fig. 3. PGA of the simulated ground-motion field in the model domain is presented by different colors. The red triangle represents the dam site. The top edges of the fault planes are indicated by the red solid lines, while the bottom edges of them are represented by the red dotted lines.

In this study, to generate the earthquake scenarios with the Monte Carlo method, all kinematic parameters of the source process (including the hypocenter, the distribution of slip, rupture time, rise time, and rake) are randomly generated in certain ranges with necessary constraints. For the source process of earthquake scenarios with, the average slip and rise time

are constrained. Specifically, the average slip is determined by the empirical earthquake source-scaling laws³⁵, and the average rise time is specified according to the empirical relation for the rise time³⁶. According to the existing research, the hypocenter of an earthquake may occur at any position of the seismic fault^{37,38}. However, due to the very limited number of earthquakes with accurate fault information derived from sufficient studies, as well as the significant randomness of the hypocenter locations, there is no well recognized conclusion regarding the spatial pattern of epicenters. Therefore, in this study, we simply assume that the hypocenters of earthquake scenarios occur randomly at any position of the seismic fault. Furthermore, it is worth emphasizing that our method is highly adaptable and can be updated to incorporate advancements in the understanding of earthquake nature, such as improved scaling laws for rupture parameters, as well as refined exploration information of faults and regional Earth model. By incorporating these advancements, our method can yield more accurate evaluations of the seismic hazard at the target site. Details of source parameters of the generated earthquake scenarios are provided in the Supplementary Information.

Reviewer #4 (Remarks to the Author):

Comment 4.1

The manuscript presents a comprehensive understanding of the significant threat posed by major earthquakes to modern society, emphasizing their destructive potential and inherent unpredictability. The concept of the maximum credible earthquake (MCE) for a specific fault, characterized by various potential scenarios with differing source processes, contributes to the complex and uncertain nature of future seismic hazards. The abstract introduces a solution in the form of a full-scenario analysis method that employs deterministic broadband simulations across a multitude of scenarios. This approach seeks to address the intricate nature of MCE hazards by considering all possible source process parameters along the fault, thereby facilitating a more comprehensive assessment.

While the abstract successfully explains this novel full-scenario analysis method and applies it to the seismic hazard assessment of China's Xiluodu dam, there appears to be potential for improvement in terms of practical application. Furthermore, the proposed method's capacity to give a variety of intensity measures, ground motion time series, and spatial ground-motion fields hold promise for a more precise and realistic evaluation of MCE hazards, with possible applications in earthquake engineering.

In conclusion, the paper presents an intriguing approach to tackling the uncertainty associated with MCE hazards and it would be appropriate to accept it for publication. Although the method's

conceptual foundation and potential contributions are commendable, a closer examination and improvement in the application phase could elevate the paper's overall impact.

Response: We express our sincere gratitude to the reviewer for the thorough review and encouraging remarks on our study, particularly the recognition of our full-scenario analysis method and its potential contributions to the assessment of MCE hazards. We agree with the reviewer's insightful and constructive comments on further examining and improving the practical application of our proposed method. Guided by the detailed comments provided, we have improved our manuscript by expanding the application of our method. A detailed point-by-point response is given as follows.

Comment 4.2

1. How does the incorporation of numerous potential scenarios with different source processes contribute to the uncertainty in predicting maximum credible earthquakes?

Response: As explained in the response to Comment 1.1, the uncertainty of earthquakes involves their highly complex physical process, including the source process, propagation path, and local site conditions. Specifically, the uncertainty in propagation path and local site conditions is associated with the velocity structures and topography, known as the epistemic uncertainty, which is expected to be improved with further refinement of the Earth model. Therefore, this study did not address the uncertainty associated with propagation path and local site conditions that related to velocity structures and topography, only focusing on the uncertainty associated with the earthquake source.

Given the current level of understandings for earthquakes, the source of potential future earthquakes is unpredictable and highly uncertain. Earthquakes can arise from different rupture process, resulting in a wide range of potential scenarios. However, by statistically analyzing the source inversion results for historical earthquakes, some empirical earthquake source-scaling laws (Thingbaijam et al., 2017), such as the fault scaling, slip amount, and rise time, are concluded for the seismic faults. Based on the recognized knowledge for earthquakes, our study considers the stochastic nature and uncertainty of the rupture process with numerous possible earthquake scenarios by constraining the rupture parameters within reasonable ranges. The seismic hazard at the target site is obtained by statistically analyzing the ground motions at the target site for all earthquake scenarios. Moreover, for the seismic hazard evaluations for major projects and densely populated urban areas, especially those prone to seismic activity, the significant faults around the considered sites usually have undergone detailed explorations, and some basic information of these faults, e.g., the approximate fault locations and fault mechanism (strike-slip or thrust), are relatively clear. By incorporating the known fault information, our method can provide more realistic and

accurate assessment for the seismic hazard of the target site.

Overall, this study acknowledges the inherent unpredictability of earthquakes and aims to move beyond a single deterministic viewpoint. By employing the full-scenario analysis method, we addressed the complex and uncertain nature of potential future earthquakes. The proposed method enables to capture the uncertainty of earthquakes and provides a more robust framework for assessing the seismic hazard by considering the actual faults, velocity structures and topography.

Comment 4.3

2. Can you elaborate on the significance of conducting deterministic broadband simulations as part of the proposed full-scenario analysis method?

Response: We agree with the reviewer that the significance of applying deterministic broadband ground-motion simulations in the proposed full-scenario analysis method should be further elaborated. Compared to the conventional seismic hazard analysis methods based on empirical ground-motion models, e.g., PSHA and DSHA, this study has several significant advantages by employing the latest deterministic broadband ground-motion simulation methods.

First, conventional seismic hazard analysis methods typically provide a limited set of seismic parameters, such as PGA (Peak Ground Acceleration) and $Sa(T1)$ (spectral acceleration at the first mode period of vibration). These simple parameters may not adequately characterize the seismic response of specific engineering structures (Luco and Cornell, 2007). In contrast, our research utilizes deterministic broadband ground-motion simulations to directly obtain the broadband ground-motion time series at the target site. This enables to assess the seismic hazard at the target site using any desired intensity measure or even broadband ground-motion time series, providing more comprehensive and accurate evaluations for specific engineering structures.

Second, broadband deterministic ground-motion simulations allow for a thorough consideration for the physical process of earthquakes from source to site, including the rupture process, propagation path, and local site conditions. Consequently, our study can take into account all factors that may influence the ground motions at the target site, e.g., the effects of rupture directivity (Somerville et al., 1997), amplification effects due to local topography (Lee, Chan, et al., 2009; Lee, Komatitsch, et al., 2009), and basin effects caused by sedimentary layers (Kawase, 2003). This is particularly significant for assessing the seismic hazard at sites adjacent to the faults where large earthquakes may occur.

Finally, by employing deterministic broadband ground-motion simulations, we can extend the analysis beyond the site-specific hazard assessment. After obtaining the seismic hazard results at the target site, we can further perform forward simulations for the obtained earthquake scenarios

corresponding to the specified hazard levels to obtain the regional spatial ground-motion field. This is of great importance for analyzing the dynamic response of large-span structures (e.g., dams, bridges, and tunnels), which can be significantly affected by the spatial heterogeneous ground motions.

Overall, the utilization of deterministic broadband ground-motion simulations offers several significant advantages over the conventional approaches. It provides the personalized information for specific structures, enables a comprehensive consideration for the realistic physical process of earthquakes, and allows for the analysis of spatial ground motion effects on large-scale structures. We believe these advantages contribute to a more robust and accurate seismic hazard analysis and have important implications in the earthquake engineering field.

Comment 4.4

3. Could you provide more insight into the specific parameters and variables considered within the full-scenario analysis approach for evaluating MCE hazards?

Response: As explained in Comment 1.1 and 4.2, in the proposed full-scenario analysis method, we focus on the uncertainties associated with the earthquake source. The source of potential future earthquakes is inherently uncertain and characterized by randomness. To obtain the seismic hazard for the target site, based on the recognized knowledge of earthquake and the explorations of the considered faults, this study comprehensively considers all uncertainties of the source of potential earthquakes, including the rupture location, hypocenter, rupture process (distributions of slip, rupture time, rise time and rake angle on the rupture fault), through massive earthquake scenarios with different source process. Furthermore, it is worth emphasizing that our method is highly adaptable and can be updated to incorporate advancements in the understanding of earthquake nature, such as improved scaling laws for rupture parameters, as well as refined exploration information of faults and regional Earth model. By incorporating these advancements, our method can yield more accurate evaluations of the seismic hazard at the target site. In line with the reviewer's suggestion, we have added more detailed descriptions of the specific parameters and variables considered in our method to the revised manuscript, as described in the response to Comment 2.4. We believe this addition will enhance comprehension of our study.

Comment 4.5

4. In what ways could the refinement of the practical application aspect enhance the practicality and adaptability of the proposed method in real-world scenarios?

Response: We appreciate the reviewer for this constructive comment regarding the refinement of

the practical application aspect for enhancing the practicality and adaptability of our method in real-world scenarios. To refine the practical application aspect of our method, we are committed to enhancing this method in the following ways:

- **Method validation:** In this study, the proposed method was validated through comparisons with empirical ground-motion models. However, further validations should be conducted through comparisons with existing seismic hazard analysis methods in practical applications. Currently, we are applying this method to evaluate the seismic hazard for several major hydraulic engineering projects in China that are located near significant faults with potential for large earthquakes. A typical example is provided in the response to Comment 4.6, and the reliability and advantages of our method are further validated in real-world scenarios.
- **Determining evaluation criteria:** In this study, much richer results are provided for the seismic hazard evaluations, including arbitrary IMs, broadband ground-motion time series, and spatial ground-motion fields. However, we need to further study the evaluation criteria for the proposed method. For instance, which IMs should be selected in seismic hazard analysis for specified engineering structures, and what hazard levels should be adopted for seismic design. Proper evaluation criteria are crucial for the application of this method in real-world scenarios.
- **Adaptability and integration:** As explained in the response to Comment 4.4, one of the strengths of our method lies in its adaptability. It can easily incorporate the latest knowledge of earthquakes and refined Earth models, leading to more accurate and reasonable seismic hazard results for the target sites. We are committed to continuously improving our method to enhance its practicality and facilitate its wider adoption in real-world applications.
- **Fault-to-structure simulation:** Current seismic response analysis for engineering structures typically assumes uniform ground-motion inputs without considering spatial heterogeneity. However, by utilizing deterministic broadband ground-motion simulations, our method can provide spatial ground-motion fields corresponding to specific seismic hazard levels through forward simulations. Therefore, our method can perfectly incorporate the newly-developed source-to-structure simulation method (Zhang et al., 2023), to achieve more realistic seismic response analysis for engineering structures by comprehensively considering the realistic physical process of earthquakes, as presented in the response to Comment 4.6.

We are actively conducting research in several ways to enhance the practicality and adaptability of this method in real-world scenarios. Our ongoing aim is to continuously refine our method, effectively addressing the challenges posed by real-world applications.

Comment 4.6

5. Considering the potential for more realistic MCE hazard evaluations, how might the proposed method influence decision-making processes in earthquake engineering projects, particularly in areas prone to seismic activity?

Response: We agree with the reviewer that the potential impacts of our method on decision-making processes in earthquake engineering projects should be further elaborated. As explained in the response to Comments 1.1 and 4.3, our study has several significant advantages over conventional approaches. The proposed method can provide much richer results for seismic hazard evaluations, and thus has great potential to improve the decision-making processes of earthquake engineering projects, particularly in areas prone to seismic activity. Here we list some potential applications of our method in the decision-making process of earthquake engineering projects.

(a) Accurate evaluations considering complete physical backgrounds

As explained in the response to Comment 4.3, our method uses deterministic broadband seismic ground-motion simulation to generate 3D ground motions for the target site. By incorporating the actual faults and Earth model into the simulations, the proposed method takes into account all factors that may influence the ground motions at the target site, e.g., the effects of rupture directivity, amplification effects due to local topography, and basin effects caused by sedimentary layers. These are of great significance for assessing the seismic hazard of engineering projects, particularly in areas prone to seismic activity where the source process has more significant effects. Additionally, our method can consider the ground motions at specific directions, which is important for engineering structures that are sensitive to specific directions of ground motions. For instance, the seismic design of gravity dams pays more attention to ground motions in the river-parallel direction.

Here, we provide an example of the application of this method in the decision-making process for selecting the site location of a high concrete gravity dam, which is currently in the planning stage in China. For this major project, preliminary exploration has identified two potential dam sites, Site A and Site B, both in close proximity to a significant fault (Fig. R3 (a)) of which the maximum magnitude is 7.5. According to the seismic hazard evaluations using PSHA for the two dam sites, the PGA of design earthquake (2% exceedance probability in 100 years) was 0.52 g for Site A, which is 5 km from the fault, and 0.525 g for Site B, which is 1.5 km from the fault. The PGA of check earthquake or MCE (1% exceedance probability in 100 years) was 0.64 g for Site A and 0.65 g for Site B. According to the seismic hazard evaluations using PSHA, despite the different distances from the fault to the two sites (5 km for Site A and 1.5 km for Site B) and the different river-parallel directions of them, the PGA values of the same hazard level are quite similar for the two sites!

Due to the lack of historical records for large earthquakes at such close proximities, stronger

spatial heterogeneity in near-field ground motion (as presented in the response to Comment 2.3), and the ergodic assumption in PSHA (Anderson and Brune, 1999), it is challenging to accurately assess the hazard from near-field major earthquakes using PSHA. Given the close proximity of the two sites, their seismic hazards might be similar for earthquakes occurring tens or even hundreds of kilometers away, assuming no significant differences of local conditions. However, in this example, for near-field earthquakes within a few kilometers, the difference in distances of the two sites from the fault (a factor of more than 3 between 1.5km and 5km) is significant. The assessment using PSHA for this project suggests a similar seismic hazard for the two sites. However, this conclusion may not be entirely accurate, as it overlooks the differences in their relative positions to the fault, local site conditions, and river-parallel directions.

Currently, we have conducted a preliminary seismic hazard analysis for the two potential dam sites using the deterministic full-scenario analysis method. Based on the results derived from our method (Fig. R4), when we overlook the distinction in ground-motion direction at the dam sites (that is, adopting the east-west direction, i.e., the river-parallel direction of Site A, uniformly for both Site A and Site B), the PGAs at 50% and 10% exceedance probabilities were 0.31 g and 0.53 g for Site A, and 0.57 g and 0.88 g for Site B, respectively. Therefore, it is revealed by our method that the seismic hazard of Site B is significantly larger than that of Site A. Furthermore, when considering the realistic river-parallel directions of different sites, at which the ground motions are more critical for the seismic hazard of gravity dams (i.e., considering ground motions of east-west for Site A and 30 degrees north of east for Site B), the seismic hazard for Site A remains unchanged, while that for Site B further increases, with PGAs at 50% and 10% exceedance probabilities reaching 0.67 g and 1.05 g, respectively. Therefore, our method demonstrated that for the project of high gravity dam, the seismic hazard of Site B is much more severe than that of Site A, rather than the similar seismic hazards derived from PSHA. This provides significant reference for the decision-making process in site selection.

Overall, for the seismic design of engineering projects prone to seismic activity, the seismic hazards at the engineering site are profoundly influenced by the actual physical backgrounds of earthquakes. Furthermore, when considering the directivity of ground motions, the ground motions at different sites close to the fault may exhibit substantial differences. Therefore, our method is promising for providing more realistic and accurate evaluations in the decision-making process for selecting the engineering sites of major projects, particularly those near significant faults.

Fig. R3 Map of the surrounding region of the considered dam sites. (a) The black triangles represent the dam sites and the red line indicates the considered fault around the dam sites; (b) The river-parallel directions of the dam site A and B are indicated by the red arrows; (c) Velocity structures around the dam sites are presented by different colors.

Fig. R4 The seismic hazards of the considered sites. (a) and (b) show the seismic hazards at the same directions for both site A and site B, and (c) presents the seismic hazards at the realistic directions for site B.

(b) Personalized evaluations for specific structures

Our method directly generates the ground-motion time series with deterministic broadband ground-motion simulations and can produce arbitrary intensity measures (IMs) for evaluating the seismic hazard. As demonstrated in Figures S5 and S6 of Supplementary Information, seismic hazards represented by different IMs may have significant differences for the same site. Therefore, by adopting the optimal IM of specific engineering structures, our method can provide personalized seismic hazards for them, thus providing more accurate and reasonable seismic hazard evaluations for earthquake engineering projects.

(c) Seismic design fully based on realistic and unified physical backgrounds

For populated urban areas and some major engineering projects with large spatial scales (e.g., dams, bridges, and tunnels), they may be significantly influenced by spatially heterogeneous ground motions, which are difficult to be considered in conventional seismic hazard analysis approaches. However, in our method, after obtaining the seismic hazard for the target site based on deterministic broadband ground-motion simulations, we can further generate spatial ground-motion fields with corresponding earthquake scenarios by forward simulations, enabling to incorporate the effects of spatially heterogeneous ground motions into the dynamic response analysis of engineering structures.

Currently, by integrating regional ground-motion simulations and structural dynamic response analysis methods, we have developed a novel source-to-structure simulation method for analyzing the dynamic response of structures under strong earthquakes. The source-to-structure simulation method was used to simulate the dynamic response of the Pacoima dam during the 1994 Northridge earthquake (Zhang et al., 2023). The simulation model and main results are shown in Figures R4, R5, and R5, and details of this simulation can refer to our recently published paper (Zhang et al., 2023). This simulation qualitatively reproduced the seismic response of Pacoima dam induced by the 1994 Northridge earthquake, e.g., the ground-motion time series on the dam body (Fig. R6) and the concrete damage distribution adjacent to the left block contraction joint (Fig. R7). Therefore, the proposed method can provide realistic ground-motion fields to achieve the source-to-structure simulation, thus enabling more realistic analysis of dynamic response for engineering structures.

As explained in the response to Comment 1.1, this study is promising for more ambitious goals to achieve the seismic design fully based on realistic physical process of earthquakes. By employing deterministic broadband ground-motion simulations that consider the actual physical backgrounds (including fault structures, velocity structures, and topography), the proposed full-scenario analysis method can directly provide arbitrary intensity measures, broadband ground-motion time series, and spatial ground-motion fields for the specified hazard levels, which can be used directly in the seismic

design of engineering structures, thereby adopting the unified earthquake scenarios with realistic physical backgrounds and specified hazard levels in both seismic hazard evaluations and structural dynamic analysis..

Fig. R5 Source-to-dam model for simulating the nonlinear dynamic response of the Pacoima dam in the 1994 Northridge earthquake. (A) Regional numerical model for simulating the ground-motion field (the red dashed frame and the star represent the rupture fault face and epicenter, respectively); (B) Finite element model for the Pacoima dam (Zhang et al., 2023).

Fig. R6 Comparison of the calculated responses (red) and records (black) at Channel 8 installed at the Pacoima dam (Zhang et al., 2023).

Fig. R7 Concrete damage distribution on Pacoima dam faces: A, downstream face, B, upstream face (Zhang et al., 2023).

Based on the deterministic broadband ground-motion simulation methods, this study achieves a deterministic full-scenario analysis for potential future earthquakes with a thorough consideration for the physical process of earthquakes. Compared with conventional approaches, this method can much richer results for seismic hazard evaluations, e.g., arbitrary IMs, broadband ground-motion time series, and spatial ground-motion fields for the specified hazard levels, enabling more realistic and accurate evaluations of seismic hazards. Furthermore, this study is promising for more ambitious goals to achieve the seismic design fully based on realistic physical process of earthquakes and holds significant implications in the field of earthquake engineering.

References:

- Anderson, J. G., 2015, The Composite Source Model for Broadband Simulations of Strong Ground Motions, *Seismological Research Letters*, 86, no. 1, 68–74, doi: 10.1785/0220140098.
- Anderson, J. G., and J. N. Brune, 1999, Probabilistic Seismic Hazard Analysis without the Ergodic Assumption, *Seismological Research Letters*, 70, no. 1, 19–28, doi: 10.1785/gssrl.70.1.19.
- Baker, J. W., N. Luco, N. A. Abrahamson, P. J. Maechling, and K. B. Olsen, 2014, ENGINEERING USES OF PHYSICS-BASED GROUND MOTION SIMULATIONS, in *Proceedings of the 10th National Conference in Earthquake Engineering Anchorage*.
- Baker, J. W., S. Rezaeian, C. A. Goulet, N. Luco, and G. Teng, 2021, A subset of CyberShake ground-motion time series for response-history analysis, *Earthquake Spectra*, 37, no. 2, 1162–1176, doi: 10.1177/8755293020981970.
- Crempien, J. G. F., and R. J. Archuleta, 2015, UCSB Method for Simulation of Broadband Ground Motion from Kinematic Earthquake Sources, *Seismological Research Letters*, 86, no. 1, 61–67, doi: 10.1785/0220140103.
- Denolle, M. A., E. M. Dunham, G. A. Prieto, and G. C. Beroza, 2014, Strong Ground Motion Prediction Using Virtual Earthquakes, *Science*, 343, no. 6169, 399–403, doi: 10.1126/science.1245678.
- Fu, H. et al., 2017, 18.9-Pflops nonlinear earthquake simulation on Sunway TaihuLight: enabling depiction of 18-Hz and 8-meter scenarios, in *Proceedings of the International Conference for High Performance Computing, Networking, Storage and Analysis Denver Colorado*, ACM, 1–12.
- Graves, R. et al., 2011, CyberShake: A Physics-Based Seismic Hazard Model for Southern California, *Pure Appl. Geophys.*, 168, nos. 3–4, 367–381, doi: 10.1007/s00024-010-0161-6.

- Graves, R. W., and A. Pitarka, 2010, Broadband Ground-Motion Simulation Using a Hybrid Approach, *Bulletin of the Seismological Society of America*, 100, no. 5A, 2095–2123, doi: 10.1785/0120100057.
- Kawase, H., 2003, 61 Site effects on strong ground motions, in *International Geophysics*, Elsevier, 1013–1030.
- Lee, S.-J., Y.-C. Chan, D. Komatitsch, B.-S. Huang, and J. Tromp, 2009, Effects of Realistic Surface Topography on Seismic Ground Motion in the Yangminshan Region of Taiwan Based Upon the Spectral-Element Method and LiDAR DTM, *Bulletin of the Seismological Society of America*, 99, no. 2A, 681–693, doi: 10.1785/0120080264.
- Lee, S.-J., D. Komatitsch, B.-S. Huang, and J. Tromp, 2009, Effects of Topography on Seismic-Wave Propagation: An Example from Northern Taiwan, *Bulletin of the Seismological Society of America*, 99, no. 1, 314–325, doi: 10.1785/0120080020.
- Luco, N., and C. A. Cornell, 2007, Structure-Specific Scalar Intensity Measures for Near-Source and Ordinary Earthquake Ground Motions, *Earthquake Spectra*, 23, no. 2, 357–392, doi: 10.1193/1.2723158.
- Maufroy, E. et al., 2015, Earthquake Ground Motion in the Mygdonian Basin, Greece: The E2VP Verification and Validation of 3D Numerical Simulation up to 4 Hz, *Bulletin of the Seismological Society of America*, 105, no. 3, 1398–1418, doi: 10.1785/0120140228.
- Milner, K. R., B. E. Shaw, C. A. Goulet, K. B. Richards-Dinger, S. Callaghan, T. H. Jordan, J. H. Dieterich, and E. H. Field, 2021, Toward Physics-Based Nonergodic PSHA: A Prototype Fully Deterministic Seismic Hazard Model for Southern California, *Bulletin of the Seismological Society of America*, 111, no. 2, 898–915, doi: 10.1785/0120200216.
- Panza, G. F., and J. Bela, 2020, NDSHA: A new paradigm for reliable seismic hazard assessment, *Engineering Geology*, 275, 105403, doi: 10.1016/j.enggeo.2019.105403.
- Pitarka, A., A. Akinici, P. De Gori, and M. Buttinelli, 2022, Deterministic 3D Ground-Motion Simulations (0–5 Hz) and Surface Topography Effects of the 30 October 2016 Mw 6.5 Norcia, Italy, Earthquake, *Bulletin of the Seismological Society of America*, 112, no. 1, 262–286, doi: 10.1785/0120210133.
- Robinson, T. R., N. J. Rosser, A. L. Densmore, K. J. Oven, S. N. Shrestha, and R. Guragain, 2018, Use of scenario ensembles for deriving seismic risk, *Proc. Natl. Acad. Sci. U.S.A.*, 115, no. 41, doi: 10.1073/pnas.1807433115.
- Rodgers, A. J., A. Pitarka, N. A. Petersson, B. Sjögreen, and D. B. McCallen, 2018, Broadband (0–4 Hz) Ground Motions for a Magnitude 7.0 Hayward Fault Earthquake With Three-Dimensional Structure and Topography, *Geophys. Res. Lett.*, 45, no. 2, 739–747, doi: 10.1002/2017GL076505.
- Somerville, P. G., N. F. Smith, R. W. Graves, and N. A. Abrahamson, 1997, Modification of

Empirical Strong Ground Motion Attenuation Relations to Include the Amplitude and Duration Effects of Rupture Directivity, *Seismological Research Letters*, 68, no. 1, 199–222, doi: 10.1785/gssrl.68.1.199.

Thingbaijam, K. K. S., P. Martin Mai, and K. Goda, 2017, New Empirical Earthquake Source-Scaling Laws, *Bulletin of the Seismological Society of America*, 107, no. 5, 2225–2246, doi: 10.1785/0120170017.

Wang, X., J. Wang, and C. Zhang, 2022, A Broadband Kinematic Source Inversion Method Considering Realistic Earth Models and Its Application to the 1992 Landers Earthquake, *JGR Solid Earth*, 127, no. 3, doi: 10.1029/2021JB023216.

Wang, X.-C., J.-T. Wang, L. Zhang, and C.-H. He, 2021, Broadband ground-motion simulations by coupling regional velocity structures with the geophysical information of specific sites, *Soil Dynamics and Earthquake Engineering*, 145, 106695, doi: 10.1016/j.soildyn.2021.106695.

Wang, X.-C., J.-T. Wang, L. Zhang, S. Li, and C.-H. Zhang, 2021, A Multidimension Source Model for Generating Broadband Ground Motions with Deterministic 3D Numerical Simulations, *Bulletin of the Seismological Society of America*, 111, no. 2, 989–1013, doi: 10.1785/0120200221.

Zhang, Y., F. Romanelli, F. Vaccari, A. Peresan, C. Jiang, Z. Wu, S. Gao, V. G. Kossobokov, and G. F. Panza, 2021, Seismic hazard maps based on Neo-deterministic Seismic Hazard Assessment for China Seismic Experimental Site and adjacent areas, *Engineering Geology*, 291, 106208, doi: 10.1016/j.enggeo.2021.106208.

Zhang, L., J.-T. Wang, Y.-J. Xu, C.-H. He, and C.-H. Zhang, 2020, A Procedure for 3D Seismic Simulation from Rupture to Structures by Coupling SEM and FEM, *Bulletin of the Seismological Society of America*, 110, no. 3, 1134–1148, doi: 10.1785/0120190289.

Zhang, M.-Z., L. Zhang, X.-C. Wang, W. Su, Y.-X. Qiu, J.-T. Wang, and C.-H. Zhang, 2023, A framework for seismic response analysis of dams using numerical source-to-structure simulation, *Earthquake Engineering & Structural Dynamics*, 52, no. 3, 593–608, doi: 10.1002/eqe.3774.

REVIEWERS' COMMENTS

Reviewer #4 (Remarks to the Author):

The paper fully addresses the suggestions made by the reviewer and can therefore be published in its current form.

Response to the comments

REVIEWER COMMENTS

Reviewer #4 (Remarks to the Author):

The paper fully addresses the suggestions made by the reviewer and can therefore be published in its current form.

Response: We sincerely appreciate the reviewer for helping improve our manuscript.